# CONSOLIDATING REINFORCEMENT LEARNING FOR MULTIMODAL DISCRETE DIFFUSION MODELS

**Tianren Ma, Mu Zhang, Yibing Wang, Qixiang Ye**[*]
University of Chinese Academy of Sciences
matianren18@mails.ucas.ac.cn  qxye@ucas.ac.cn

## ABSTRACT

Optimizing discrete diffusion model (DDM) with rewards remains a challenge—the non-autoregressive paradigm makes importance sampling intractable and rollout complex, puzzling reinforcement learning methods such as Group Relative Policy Optimization (GRPO). In this study, we introduce **MaskGRPO**, the first viable approach to enable scalable multimodal reinforcement learning in discrete diffusion with effective importance sampling and modality-specific adaptations. To this end, we first clarify the theoretical foundation for DDMs, which facilitates building an importance estimator that captures valuable token fluctuation for gradient updates. We then delicately tailored the rollout method for visual sequences, which yields diverse completions and reliable optimization gradients. Across math reasoning, coding, and visual generation benchmarks, MaskGRPO brings more stable and efficient updates, **doubling** reinforcement learning gains while speeding up training by up to **30%**. This study establishes MaskGRPO as a systematic policy optimization approach and the first practical way for discretized visual diffusion. The code is available at https://github.com/martian422/MaskGRPO.

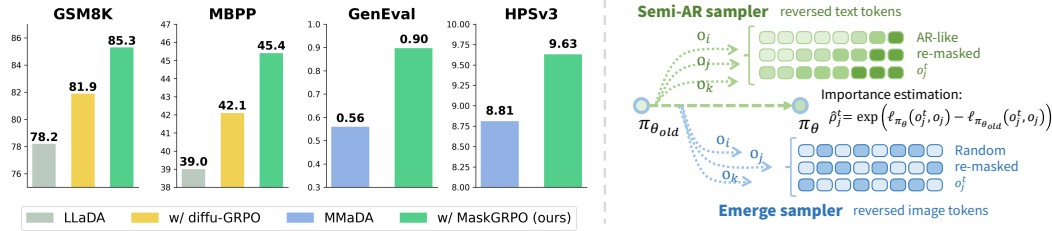

Figure 1: Left: **MaskGRPO** consistently improves the base model with significant RL income across text and image generation tasks. Right: an intuitive demonstration of our method, integrated with modality-specific innovations on importance estimation and sampling methods.

## 1 INTRODUCTION

Recent progress of post-training generative models has been driven by advances in optimization algorithms, architectural design, and large-scale reward-based learning (Rafailov et al., 2023; Liu et al., 2025; Deng et al., 2025). Among these, Group Relative Policy Optimization (GRPO) (Shao et al., 2024) has emerged as a powerful and scalable paradigm, improving reasoning performance of large language models and enhancing preference alignment of visual generative models. However, extending such policy optimization to discrete diffusion models (DDMs) remains a challenge.

Unlike autoregressive models that decode sequentially, discrete diffusion generates tokens in parallel at arbitrary positions. This parallelism complicates both *rollout generation*, where stochastic yet coherent samples are required (Liu et al., 2025) for exploration, and *importance estimation*, which is crucial for optimization (Schulman et al., 2017). Existing approaches offer only partial solutions: semi-autoregressive samplers (Arriola et al., 2025; Nie et al., 2025) mitigate inference issues for text,

---

[*]Corresponding author.

while low-confidence re-masking for images (Chang et al., 2022) lack stochastic flexibility for robust group comparisons. Similarly, early attempts Zhao et al. (2025); Tang et al. (2025) at importance estimation relied on masking heuristics that violated conditioning assumptions. Monte Carlo–based estimators Zhu et al. (2025) improve faithfulness but remain computationally expensive.

In this study, we propose **MaskGRPO**, a consolidated extension of GRPO to multimodal discrete diffusion (shown in Fig. 1), built upon modality-specific innovations in both sampling and importance estimation. MaskGRPO is guided by the distinct structural properties of language and vision:

**Language.** While training native discrete diffusion models depart from the autoregressive[1] paradigm, their prediction on natural language still exhibits a degree of "ARness" (Gong et al., 2025): tokens closer to observed context are predicted with higher certainty, and rollouts diverse as length extends. Leveraging this property, we introduce a fading-out masking estimator, which progressively increases the masking rate toward later tokens with well-controlled randomness. This concentrates estimation on high-uncertainty regions, towards a more efficient and empirically reliable objective.

**Vision.** Images lack a sequential structure and exhibit strong global token correlations (Chan et al., 2024). We argue that effective likelihood estimation requires highly truncated mask rates to capture informative variation. Furthermore, we adapt a sampler that relaxes rigid scheduling constraints in existing methods via probabilistic decoding. By encouraging diverse yet high-quality rollouts, the training process better aligns with the GRPO principle of exploiting group-relative advantages.

Beyond empirical results in mathematical reasoning and coding that almost double the income from RL, our method also demonstrates significant improvement on text-image alignment, and visual fidelity. Building upon a clarified foundation for DDMs, our analysis highlights that policy optimization in discrete diffusion is only effective when samplers and estimators are designed in a modality-aware fashion. Through these contributions, we build the first systematic GRPO approach for multimodal discrete diffusion. This establishes a new foundation for reward-based learning in DDMs and points toward a more general theory of preference-driven optimization across modalities.

## 2 PRELIMINARIES

### 2.1 DISCRETE DIFFUSION MODEL

DDM defines a forward process on discrete variables by gradually corrupting tokens to absorbing state $\mathbf{m}$ through a continuous-time Markov process. We denote clean data as $x_{t=0}$ ($x_0$ for short), and noise it gradually as $t \to 1$. Let $\alpha_t$ be the noise scheduler (a monotonically decreasing survival function that satisfies $\alpha_0 = 1, \alpha_1 = 0$ ), the corrupted data distribution at time $t$ is determined as

$$x_t \sim q(x_t|x_0, t), q(x_t|x_0, t) = \text{Cat}(x_t; \alpha_t x_0 + (1 - \alpha_t)\mathbf{m}) \tag{1}$$

Let $\delta(x_{(t,i)}, \mathbf{m})$ be the indicator function that is activated only if the $i$-th position of $x_t$ is $\mathbf{m}$. For a linear scheduler, the objective is derived as the evidence lower bound (ELBO) of $\log \pi_\theta(x_0|x_t)$:

$$\mathcal{L}_{\text{DDM}} = -\mathbb{E}_{t, x_0, x_t}\Big[\frac{1}{t} \sum_{i=1}^{L} \delta(x_{(t,i)}, \mathbf{m}) \log \pi_\theta(x_{(0,i)}|x_t)\Big] = -\mathbb{E}_{t, x_0, x_t}[\ell_{\pi_\theta}(x_t, x_0)]. \tag{2}$$

We denote the loss term by $\ell_{\pi_\theta}(x_t, x_0)$ for later usage. For conditional generation where a prompt $\mathbf{c}$ is given, we write $\ell_{\pi_\theta}(x_t, x_0|\mathbf{c})$ for simplicity. Following MDLM's deduction Sahoo et al. (2024), assume that the network can reconstruct $x_0$ perfectly, we use $\pi_\theta(x_t)$ to approximate this denoising process, and get the sampling rule as

$$p_\theta(x_s|x_t) = \begin{cases} 1, & \text{if } x_s = x_t, \ x_t \neq \mathbf{m}, \\ \frac{1-\alpha_s}{1-\alpha_t}, & \text{if } x_s = \mathbf{m}, \ x_t = \mathbf{m}, \\ \frac{\alpha_s - \alpha_t}{1-\alpha_t}\pi_\theta(x_t), & \text{if } x_s \neq \mathbf{m}, \ x_t = \mathbf{m}, \\ 0, & \text{otherwise.} \end{cases} \tag{3}$$

---

[1]In this paper, we use *autoregressive* in its conventional sense of causally ordered next-token prediction.

## 2.2 GRPO FOR AUTOREGRESSIVE MODEL

Formally, for each question $\mathbf{c} \sim \mathcal{D}$, GRPO samples a group of $G$ responses (also addressed as rollouts) $\{o_1, o_2, \ldots, o_G\}$ from the old policy $\pi_{\theta_{\text{old}}}$. For rollout $o_i$, the reward system gives a action value $r_i$, and the relative advantage of it is normalized as

$$A_i = \frac{r_i - \text{mean}(\{r_j\}_{j=1}^G)}{\text{std}(\{r_j\}_{j=1}^G)}. \tag{4}$$

For position $1 \leq k \leq |o_i|$, the token-level importance is calculated as

$$\rho_i^k = \frac{\pi_\theta(o_i^k|\mathbf{c}, o_i^{<k})}{\pi_{\theta_{\text{old}}}(o_i^k|\mathbf{c}, o_i^{<k})} = \exp\left(\log \pi_\theta(o_i^k|\mathbf{c}, o_i^{<k}) - \log \pi_{\theta_{\text{old}}}(o_i^k|\mathbf{c}, o_i^{<k})\right). \tag{5}$$

With $\epsilon$ controlling the clip range, the reward component is defined as

$$R(\theta, \mathbf{c}) = \frac{1}{G} \sum_{i=1}^G \frac{1}{|o_i|} \sum_{k=1}^{|o_i|} \min\left(\rho_i^k A_i, \text{clip}(\rho_i^k, 1-\epsilon, 1+\epsilon)A_i\right), \tag{6}$$

Finally, the GRPO objective is expressed as a reward-penalty tradeoff, as

$$\max_\theta \mathbb{E}_{\mathbf{c}\sim\mathcal{D}, o_{1:G}\sim\pi_\theta(\cdot|\mathbf{c})}\left[R(\theta, \mathbf{c}) - \beta \mathbb{D}_{\text{KL}}\left[\pi_\theta(\cdot|\mathbf{c}) \| \pi_{\text{ref}}(\cdot|\mathbf{c})\right]\right], \tag{7}$$

where $\beta$ regulates the strength of the KL regularization.

## 2.3 ACCOMMODATING GRPO FOR DISCRETE DIFFUSION MODEL

We accommodate GRPO's objective for DDM in this section. To avoid confusion and align with the settings in Eq. 7, we move the timestep notation to the top-right corner of the variable. Accordingly, let $o^t \sim q(o^t|o, t)$ denote the corrupted (reversed) response $o$ with strength $t$. Rolling back sequentially on AR model's response can be regarded as reversing on the timeline of $o$, *i.e.*, the first $k$ tokens $o^{\leq k}$ of AR model's response can be regarded as a re-masked $o$ with $t = \frac{|o_i|-k}{|o_i|}$. Hence, for each completion, we can approximate the sub-sequence level importance $\rho^t$ by gradually reversing it.

Recalling $\ell_{\pi_\theta}$ from Eq. 2, for a small interval $\delta t$, let $\dot{o}^t = o^t - o^{t+\delta t}$ denote the tokens that are unmasked from timestep $t + \delta t$ to $t$. We propose that the differentiation on DDM's intractable log-likelihood can be approximated[2] using

$$\log \pi_1(\dot{o}^t|\mathbf{c}, o^{t+\delta t}) - \log \pi_2(\dot{o}^t|\mathbf{c}, o^{t+\delta t}) \approx \ell_{\pi_1}(o^t, o|\mathbf{c}) - \ell_{\pi_2}(o^t, o|\mathbf{c}) \tag{8}$$

The above expression indicates that, to evaluate the fluctuation of likelihood for newly unmasked tokens in $o^t$, we can utilize the difference of model's prediction for the full sequence at time $t$. Hence, we derive the calculable importance estimation and KL divergence as

$$\hat{\rho}_i^t = \exp\left(\ell_{\pi_\theta}(o_i^t, o_i|\mathbf{c}) - \ell_{\pi_{\theta_{\text{old}}}}(o_i^t, o_i|\mathbf{c})\right). \tag{9}$$

$$\hat{\mathbb{D}}_{\text{KL}}^{i,t} = \exp\left(\ell_{\pi_{\theta_{\text{ref}}}}(o_i^t, o_i|\mathbf{c}) - \ell_{\pi_\theta}(o_i^t, o_i|\mathbf{c})\right) - \left(\ell_{\pi_{\theta_{\text{ref}}}}(o_i^t, o_i|\mathbf{c}) - \ell_{\pi_\theta}(o_i^t, o_i|\mathbf{c})\right) - 1. \tag{10}$$

We temporarily skip the clip operation for simplicity and accommodate Eq. 7 as

$$\max_\theta \mathbb{E}_{\mathbf{c}\sim\mathcal{D}, o_{1:G}\sim\pi_\theta(\cdot|\mathbf{c})}\left[\frac{1}{G}\sum_{i=1}^G \frac{A_i}{|o_i|}\sum_{j=1}^\mu (\hat{\rho}_i^{t_j} - \beta\hat{\mathbb{D}}_{\text{KL}}^{i,t_j})\right], \quad t_j = j/\mu \tag{11}$$

Upon this foundation, we revisit the prior endeavors on DDM optimization: *diffu*-GRPO Zhao et al. (2025) applies masks to prompts and extract likelihood on the entirely masked completions $o^{t=1}$. Following LLaDA-1.5 Zhu et al. (2025), UniGRPO Yang et al. (2025) iteratively masks varying ratio of completions. While these strategies provide gradient signals, they either disrupt the conditional dependency or pose high budget for Monte Carlo style estimations. In summary, current inefficiency of likelihood estimation ties DDM to limited settings and obscures its potential in broader contexts, especially in reasoning or visual generation that may involve thousands of tokens per sample.

---

[2]While previous works omit correlating discussion, this approximation is critical for deduction and code implementation. Please refer to the Appendix C for details.

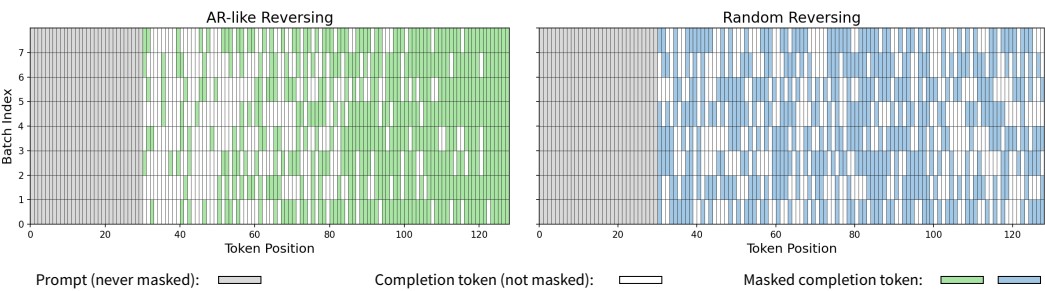

Figure 2: A demonstration of different reversing (re-mask) methods with ratio $r = 0.6$. Random reversing (right) applies masks to all the tokens with equal probability, while AR-like reversing (left) adapts a fading-out strategy. See Appendix F for complete showcases.

# 3 MASKGRPO

## 3.1 IMPORTANCE ESTIMATION

Importance sampling is central to the GRPO objective, where it functions as an effective "reweighting" of rewards to align with the current policy's distribution with reduced bias. In practice, the importance is calculated by the differentiate of predictions as in Eq. 8. Inspired by model rollouts' behavior, we modify the estimator to capture valuable fluctuation instead of highly confident predictions from low-mask sequences.

First, to allocate the timestep budget effectively, we clamp the sampling range from $(0, 1)$ to $(\gamma, 1)$, where $\gamma$ serves as a cut-off of mask ratios. Second, rather than relying on randomly masking, we design estimators that utilize the autoregressive biases in language and locality-driven correlations in vision. We implement the reverse process by managing independent random seeds on each device, which is crucial for stable importance and KL computation. With the designed operator, we obtain stable and low-variance estimates (see 4.3 for details) driven by the stochasticity of $\hat{o}_t \sim \text{Rev}(o, t)$.

**Let language tokens fade out.** DDMs exhibit a causal bias for language (Gong et al., 2025), particularly in logically related tasks such as math and code. This property, referred to as AR-ness, has been identified to have a strong correlation with model's overall performance. Besides, as the semi-autogressive sampler (Alg. 3) is utilized, the rollouts also exhibit higher divergence as block extends, *i.e.*, at the start of response, the model's reasoning are rather simple setups, while real divergence emerges as the reasoning proceeds (see D.1 for evidence). This observation motivates us to exploit the importance estimation through an AR-like reversing procedure, and assign higher attention to the later tokens. Alg. 1 maintains a delicate balance between *randomness* and *fading-out* property with no additional calculation, and serves as a plug-and-play module, which can bring significant improvement by simply replacing the original reverse method with our AR-like version.

| **Algorithm 1** AR-like Reversing (for text, ours) | **Algorithm 2** Random Reversing (for image) |
|---|---|
| **Require:** Token batch $X \in \mathbb{R}^{B \times L}$, prompt indicator $C$, mask token $\mathbf{m}$, seed $s$, ratio $r$ | **Require:** Token batch $X \in \mathbb{R}^{B \times L}$, prompt indicator $C$, mask token $\mathbf{m}$, seed $s$, ratio $r$ |
| 1: Set random seed with $s$ | `# randomness is managed` |
| 2: Prompt (padded) length $L_c \leftarrow \sum C$ | 1: Set random seed with $s$ |
| 3: Non-prompt length $L_o \leftarrow L - L_c$ | `# similar as in` $q(x_t\|x_0, t)$ |
| 4: Linear decay $d \leftarrow \text{linspace}(1, 0, L_o)$ | 2: Constant curve $p \leftarrow r^L$ |
| 5: Normalize $p_n \leftarrow \frac{d \cdot ((1-r)L_o)}{\sum d}$ | 3: Expand to batch size $P \leftarrow \text{repeat}(p, B)$ |
| 6: $p \leftarrow 0^{L_c} \oplus p_n, P \leftarrow \text{repeat}(p, B)$ | 4: Sample random matrix $R \sim U(0, 1)^{B \times L}$ |
| 7: $R \sim U(0, 1)^{B \times L}, M \leftarrow (\neg C) \wedge (R > P)$ | 5: Determine mask $M \leftarrow (\neg C) \wedge (R < P)$ |
| 8: Apply masking $\tilde{X} \leftarrow \text{where}(M, \mathbf{m}, X)$ | 6: Apply masking $\tilde{X} \leftarrow \text{where}(M, \mathbf{m}, X)$ |
| 9: **return** $\tilde{X}, M$ | 7: **return** $\tilde{X}, M$ |

## 3.2 ROLLOUT ADAPTION

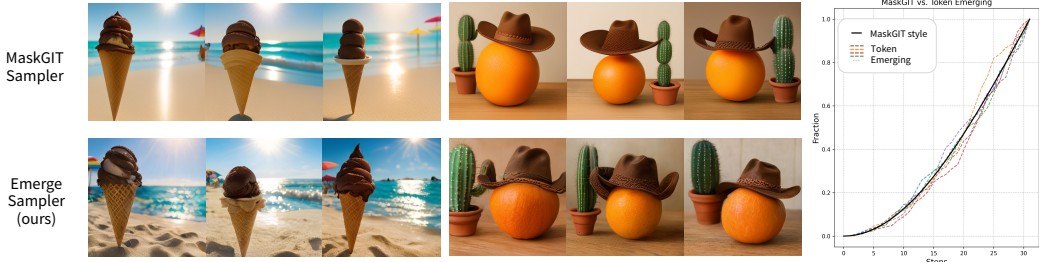

Figure 3: A comparison of MaskGIT-style sampler and emerge sampler. With identical sampling parameters on MMaDA (equipped with a 8192-vocab visual tokenizer Xie et al. (2024)), images sampled by the emerge method demonstrate better texture and expressiveness.

**Let visual tokens emerge from masks.** Complementing the estimators, we align the rollout sampler with structural patterns, enabling efficient and stable training with GRPO. For text generation, we employ the widely adopted semi-autoregressive sampler (Alg. 3), which uses a low-confidence re-mask strategy with block-wise decoding, reflecting the inherently sequential structure of language.

While expressed as discrete vocabularies, visual tokens differ from language significantly in terms of entropy, bits of information and so on (Chan et al., 2024). The confidence-based MaskGIT sampler, which was proposed with a 1024-vocab tokenizer, while viable, does not perform as expected for high-fidelity tokenizers. This curse has been partly discussed as sampling inaccuracy (Zheng et al., 2025; Ma et al., 2025a), yet we noticed that it becomes severe on large-vocabulary visual tokenizers and cannot be addressed by simply operating at higher precision. To overcome this problem which hinders model's potential on visual generation, we refer to Sahoo et al. (2024); Shi et al. (2024) and adapt their sample intuitiveness to discrete image generation. As shown in Fig 3 and Alg. 4, the emerge sampler does not enforce a decoding quantity per prediction, but let the visual tokens emerge from masks naturally with probabilistic control. While faithful to the principled DDM theory (Eq. 3), the emerge method shows significantly better expressiveness for vision. $\mathbf{Cat}(\cdot, \pi)$ is implemented with `torch.multinomial`.

As for reversing, unlike continuous diffusion RL methods Liu et al. (2025) where the importance is calculated across almost all traversed timesteps, we find that discrete visual diffusion requires a large truncation on reverse range. The tokenized patches show strong global correlations, making the prediction largely insensitive to small mask ratios, and small truncation may even lead to exploded variance. Therefore, we keep the reversing random as Alg. 2, while the reverse strength is held at high level, *e.g.,* setting $\gamma = 0.8$ to obtain meaningful importance estimates.

---

**Algorithm 3** Semi-autoregressive (for text)

1: **Inputs:** prompt $c$, completion length $L$
2: **Settings:** block length $L_{\text{block}}$, token per step $s$
3: Initialize: $x \leftarrow \mathbf{m}^L$.
4: **for** $b = 1$ to $L/L_{block}$ **do**
5:      range $\leftarrow [(b-1)L_{\text{block}}, bL_{\text{block}}]$
6:      $x_b \leftarrow x[\text{range}]$
7:      **for** $k = 1$ to $L_{\text{block}}/s$ **do**
8:          $p_{\text{conf}} \leftarrow f_\theta(x, c)[\text{range}]$
9:          $x_b \leftarrow \text{where}(x_b = \mathbf{m}, \text{argmax}(p_{\text{conf}}), x_b)$
10:          $m_{\text{re}} \leftarrow \text{argsort}(p_{\text{score}})[: L_{\text{block}} - ks]$
11:          $x_b \leftarrow \text{where}(m_{\text{re}}, \mathbf{m}, x_b))$
12:      **end for**
13:      $x[\text{range}] \leftarrow x_b$
14: **end for**
15: **Return:** fully unmasked sequence $x$

**Algorithm 4** Token Emerge (for image, ours)

1: **Inputs:** prompt $c$, completion length $L$
2: **Settings:** total steps $K$, scheduler $\alpha_t$
3: Initialize: $x_1 \leftarrow \mathbf{m}^L$.
4: **for** $k = 1$ to $K$ **do**
5:      $t \leftarrow \frac{K-k+1}{K}, s \leftarrow \frac{K-k}{K}$
6:      $\text{logits}_{\text{c}} \leftarrow f_\theta(x_t, c)$
7:      $\text{logits}_{\text{u}} \leftarrow f_\theta(x_t, \varnothing)$
8:      $\text{logits} \leftarrow \text{logits}_{\text{c}} + w \cdot (\text{logits}_{\text{c}} - \text{logits}_{\text{u}})$
9:      $\pi \leftarrow \text{Softmax}(\text{logits})$
10:      $q_s \leftarrow \frac{\alpha_s - \alpha_t}{1 - \alpha_t} \cdot \pi + \delta_{\mathbf{m}} \cdot \frac{1 - \alpha_s}{1 - \alpha_t}$
11:      $x_{\text{pred}} \leftarrow \hat{x} \sim \mathbf{Cat}(\hat{x}; q_s)$
12:      $x_s \leftarrow \text{where}(x_t = \mathbf{m}, x_{\text{pred}}, x_t)$
13: **end for**
14: **Return:** fully unmasked sequence $x_0$

## 3.3 ALGORITHM FRAMEWORK

To summarize, we implement GRPO for discrete diffusion models with integrated efficient modality-specific importance estimators, and modification on visual sequence sampling. The overall framework consists of: (*i*) **Sample** a set of full completions for each prompt **c** with method 3, 4, (*ii*) for each completion $o$, generate multiple masked $\hat{o}_t \sim \mathbf{Rev}(o, t)$, where $\mathbf{Rev}(\cdot, t)$ is our designed reverse function with controllable randomness, as shown in Alg. 1, 2. (*iii*) estimating per-completion advantages $A$ based on reward $r$, and (*iv*) updating the policy using importance $\hat{\rho}^{t_j}$ and divergence $\hat{\mathbb{D}}_{\mathrm{KL}}^{t_j}$. An algorithmic demonstration of MaskGRPO is provided in Alg. 5.

---

**Algorithm 5** MaskGRPO Policy Gradient Optimization (ours)

---

**Require:** Reference model $\pi_{\mathrm{ref}}$, prompt distribution $\mathcal{D}$, number of completions per prompt $G$, number of inner updates $\mu$, random seed set $\mathcal{S}_{j=1\sim\mu}$,
1: Initialize policy $\pi_\theta \leftarrow \pi_{\mathrm{ref}}$
2: **while** not converged **do**
3:      $\pi_{\mathrm{old}} \leftarrow \pi_\theta$
4:      Sample a prompt $\mathbf{c} \sim \mathcal{D}$
5:      **Sample** $G$ completions $o_i \sim \pi_{\mathrm{old}}(\cdot \mid \mathbf{c}), \ i \in [G]$
6:      For each $o_i$, compute reward $r_i$ and advantage $A_i$ using Eq. 4
7:      **for** gradient update iterations $j = 1, \ldots, \mu$ **do**
8:          Get timestep: $t_j \leftarrow \gamma + (1 - \gamma)\frac{j}{\mu}$
9:          Construct masked completion $\hat{o}_{i,t_j} \sim \mathbf{Rev}(o_i, t_j, \mathcal{S}_j)$
10:         For $\pi_\theta, \pi_{\mathrm{old}}, \pi_{\mathrm{ref}}$, use Eq. 9, 10 to estimate importance $\hat{\rho}_i^{t_j}$ and $\hat{\mathbb{D}}_{\mathrm{KL}}^{i,t_j}$ with $\hat{o}_{i,t_j}$
11:         Compute MaskGRPO objective in Eq. 11 and update $\pi_\theta$ via gradient descent
12:      **end for**
13: **end while**
14: **return** $\pi_\theta$

---

# 4 EXPERIMENT

## 4.1 SETUP

We conduct experiments across multimodal scenarios and evaluate MaskGRPO extensively on math reasoning, coding, and text-to-image generation benchmarks. In this section, we provide a brief demonstration with training and evaluation details in Appendix. E.

**Models**    We select LLaDA-8B-Instruct Nie et al. (2025), an open-sourced native DDM and its multimodal adaption MMaDA-8B-Base Yang et al. (2025), which unlocks the ability to perform discrete diffusion on image sequences, as the start point of optimization. Both models are initialized from publicly available pre-trained checkpoints.

**Reward function**    We follow common practices and use a reward system for reinforcement learning. For language tasks, we utilize a simple composed reward function of formatting and correctness. For image generation, following the practice of recent RL work Geng et al. (2025), the reward is composed with UnifiedReward Wang et al. (2025a), for scoring text-image alignment, HPSv3 Ma et al. (2025b), for assessing the aesthetic quality of the image alongside its alignment, and the classic CLIP Score Hessel et al. (2022).

**Metrics**    We evaluate the proposed MaskGRPO framework on text and image generation tasks, using a suite of standard benchmarks to assess its performance gain. (1) **Text Generation**. To evaluate model's mathematical reasoning and coding capabilities, we use GSM8K Cobbe et al. (2021), MATH500 Lightman et al. (2023) and MBPP Austin et al. (2021) benchmarks. We also compare with the recent RL baselines including *diffu*-GRPO Zhao et al. (2025), wd1 Tang et al. (2025) and UniGRPO (re-implemented due to unavailable codebase) Yang et al. (2025) on these tasks. (2) **Image Generation**. To evaluate model's text-image alignment, we first utilize the widely adopted

Table 1: **Evaluation on math reasoning and coding benchmarks.** For fair comparison, we choose LLaDA-8B-Instruct as the initial point. All results are reported with Pass@1 metric. [†] refers to our re-implementation.

| RL Method / Seq Len | GSM8K | | MATH500 | | MBPP |
|---|---|---|---|---|---|
| | **256** | **512** | **256** | **512** | **256** |
| LLaDA-8B-Instruct | 76.7 | 78.2 | 32.4 | 36.2 | 39.0 |
| w/ *diffu*-GRPO Zhao et al. (2025) | 79.8 (+3.1) | 81.9 (+3.7) | 34.4 (+2.0) | 39.0 (+2.8) | 42.1 (+3.1) |
| w/ UniGRPO[†] Yang et al. (2025) | 81.1 (+4.4) | 82.0 (+3.8) | 35.0 (+2.6) | 38.8 (+2.6) | 43.1 (+4.1) |
| w/ `wd1` Tang et al. (2025) | 80.8 (+4.1) | 82.3 (+4.1) | 34.4 (+2.0) | 39.0 (+2.8) | – |
| w/ TraceRL[†] Wang et al. (2025b) | 82.1 (+5.4) | 83.3 (+5.1) | 35.9 (+3.5) | 39.5 (+3.3) | 43.9 (+4.9) |
| w/ MaskGRPO (ours) | **84.7 (+8.0)** | **85.6 (+7.4)** | **37.6 (+5.2)** | **41.5 (+5.3)** | **45.4 (+6.4)** |

Table 2: **Evaluation on GenEval.** SFT indicates that we SFT the base model with BLIP3-o dataset Chen et al. (2025a) for clean instruction-tuning data distilled from GPT-4o.

| Model | GenEval↑ | | | | | | |
|---|---|---|---|---|---|---|---|
| | **Single.** | **Two.** | **Count.** | **Color.** | **Pos.** | **Attr.** | **Overall** |
| *Continuous Generation* | | | | | | | |
| SDXL Podell et al. (2023) | 0.98 | 0.74 | 0.39 | 0.85 | 0.15 | 0.23 | 0.55 |
| DALL-E 3 Betker et al. (2023) | 0.96 | 0.87 | 0.47 | 0.83 | 0.43 | 0.45 | 0.67 |
| SD3.5-L Esser et al. (2024) | 0.98 | 0.89 | 0.73 | 0.83 | 0.34 | 0.47 | 0.71 |
| FLUX.1-dev Labs (2025) | 0.98 | 0.93 | 0.75 | 0.93 | 0.68 | 0.65 | 0.82 |
| *Discrete Generation* | | | | | | | |
| Show-o Xie et al. (2024) | 0.95 | 0.52 | 0.49 | 0.82 | 0.11 | 0.28 | 0.53 |
| Janus-Pro Chen et al. (2025b) | 0.99 | 0.89 | 0.59 | 0.90 | 0.79 | 0.66 | 0.80 |
| MMaDA Yang et al. (2025) | 0.96 | 0.60 | 0.45 | 0.81 | 0.14 | 0.25 | 0.56 |
| w/ UniGRPO Yang et al. (2025) | 0.99 | 0.76 | 0.61 | 0.84 | 0.20 | 0.37 | 0.63 |
| w/ MaskGRPO (ours) | 0.99 | 0.90 | 0.75 | 0.88 | 0.65 | 0.69 | **0.81** |
| w/ SFT+MaskGRPO (ours) | 0.99 | 0.92 | 0.85 | 0.91 | 0.91 | 0.83 | **0.90** |

GenEval Ghosh et al. (2023) and DPG-Bench Hu et al. (2024) (see Appendix F) as the metrics. Then, we evaluate the generated samples' aesthetic quality using human preference scorers like DeQA You et al. (2025), ImageReward Xu et al. (2023), and HPSv3 Ma et al. (2025b). For references rather than definitive comparisons, we include both specialized diffusion models, such as SDXL, and leading discrete generation models like Show-o Xie et al. (2024) and Janus-Pro Chen et al. (2025b).

## 4.2 PERFORMANCE AND COMPARISON RESULTS

**Language Tasks** MaskGRPO substantially enhances the mathematical reasoning and coding capabilities of LLaDA. As shown in Table 1, our method achieves over 5% absolute improvement in accuracy on GSM8K, MATH500, and MBPP, nearly doubling the RL gains compared to prior methods with less steps (6000 vs 7000+). On GSM8K, MaskGRPO allows the model to surpass previous approaches while requiring only half the completion length (256 vs 512), demonstrating its effective improvement in reasoning ability. Representative examples are included in Appendix F.3.

**Visual Generation Tasks** To our knowledge, MaskGRPO is the first method to achieve effective GRPO optimization of aesthetic quality and text–image alignment in discrete diffusion models. Table 3 shows consistent improvements in alignment with human preferences, which are not reported in previous DDM works. Moreover, results in GenEval (Table 2) further confirm the effectiveness of our framework: with a well-designed RL setup, discrete generation models can approach the performance of leading commercial systems. Qualitative samples are provided in Fig. 4. Notably, while UniGRPO claims its mastery of visual RL, neither training configuration nor performance comparison is reported. Therefore, we index its final performance as RL results.

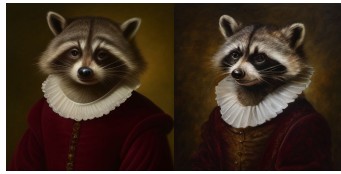 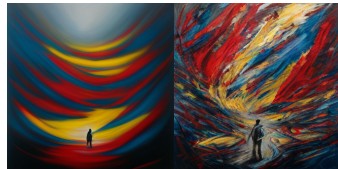 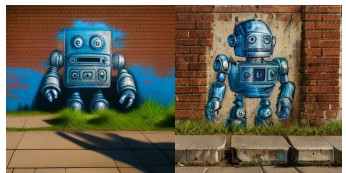

An exquisite oil painting that captures a raccoon with an almost human-like poise, dressed in attire reminiscent of the 17th century. The raccoon's fur is rendered in rich, textured strokes of brown and gray, and it wears a white ruffled collar and a deep red velvet coat that would befit a noble of Rembrandt's era. The background of the painting is a muted blend of dark, warm tones, creating a subtle contrast that draws attention to the subject's detailed and expressive face.

An abstract oil painting that depicts a chaotic blend of vibrant colors and swirling patterns, giving the impression of a vast, disorienting landscape. The canvas is filled with bold strokes of reds, blues, and yellows that seem to clash and compete for space, symbolizing the complexity and confusion of navigating through life. Amidst the turmoil, a small, indistinct figure appears to be wandering, searching for direction in the overwhelming expanse.

A vibrant depiction of a robot, spray-painted in hues of blue and silver, adorns an aged brick wall. The sidewalk in front of the wall, made of weathered concrete slabs, is interrupted by tufts of green grass sprouting from the cracks. The artwork casts a shadow on the uneven ground, hinting at the late afternoon sun.

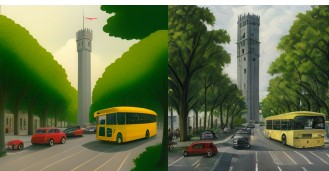 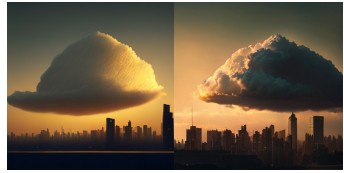 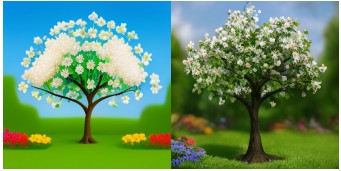

A tall, gray tower looms over the bustling street below, where cars and buses navigate through the flow of traffic. The street is canopied by a row of leafy green trees, which cast dappled shadows onto the asphalt. Behind a ruddy red car parked along the side of the road, more trees with thick foliage provide a backdrop of natural green against the urban environment. A large yellow bus makes its way down the lane, adding vibrancy to the cityscape

A tranquil cityscape with high-rise buildings silhouetted against the evening sky. In the foreground, a large, fluffy, solitary cloud hovers subtly, its edges tinged with a golden hue from the setting sun. Below the cloud, in elegant, rounded cursive letters, the words 'contemplate the clouds' invite onlookers to pause and reflect amidst the urban environment.

A picturesque scene featuring a small tree, its branches laden with delicate white blossoms, standing in the center of a lush green lawn. the tree's rounded shape is accentuated by the contrast of the vibrant green leaves against the pure white petals. surrounding the tree, a variety of colorful flowers can be seen, adding to the charm of the tranquil setting.

Figure 4: **Qualitative comparison.** Results are generated with identical sampling parameters and shown in {*original, w/ RL*} pairs. MaskGRPO demonstrates substantial improvement on the aesthetic quality of generated images, in terms of artistic style, photographic details and overall atmosphere. We strongly recommend that the readers view more portrait samples at Fig. 10.

Table 3: **Evaluation on compositional generation and human preference metrics.** We calculate the Preference Scores on samples generated by DPG-Bench prompts.

| Model | Compositional Generation | | Preference Scores | | |
|---|---|---|---|---|---|
| | GenEval | DPG-Bench | DeQA | ImageReward | HPSv3 |
| MMaDA | 0.56 | 0.71 | 3.99 | 0.93 | 8.81 |
| w/ MaskGRPO | 0.81 | 0.75 | 4.10 | 1.18 | 9.40 |
| w/ SFT+MaskGRPO | 0.90 | 0.82 | 4.18 | 1.30 | 9.63 |

## 4.3 DISCUSSION

**Truncation hyper-parameters** We perform ablation studies on GSM8K with timestep truncation ratios $\gamma \in 0.2, 0.4, 0.6, 0.8$ for 4000 steps. As shown in Fig. 5 (*a*), both the absence of truncation and overly aggressive truncation degrade training stability. To promote stable learning rather than premature convergence, we adopt $\gamma = 0.6$ as the default setting.

**Reverse method** Concurrent work TraceRL Wang et al. (2025b) proposes to track the generation trace and reverses strictly along these recorded traces. This mechanism relies on predefined paths and has only been demonstrated on block-attention architecture (SDAR JetAstra-ML (2025)). Moreover, TraceRL requires maintaining trace maps throughout training, and its deterministic reversal leads to limited flexibility in estimating prior tokens. We re-implement TraceRL on the full-attention language model, namely LLaDA-8B-Instruct, and report results in Tab. 1. In addition, we fix $\gamma = 0.6$ and ablate the reverse strategies in Fig. 5 (*b*). Our proposed AR-like reversing method consistently outperforms TraceRL in reinforcement learning. We attribute this performance gap to TraceRL's path-dependent formulation, which constrains exploration and induces biased estimation of sequence-level importance. For further qualitative evidence, we provide a visualized comparison of reversing strategies under varying ratios in Appendix F.1.

**KL divergence comparison** We also present the evolution of KL divergence during RL training across different reverse strategies in Fig. 5 bottom. A gradual increase in divergence is typical in GRPO-style optimization, reflecting the model's shift away from the reference distribution. For

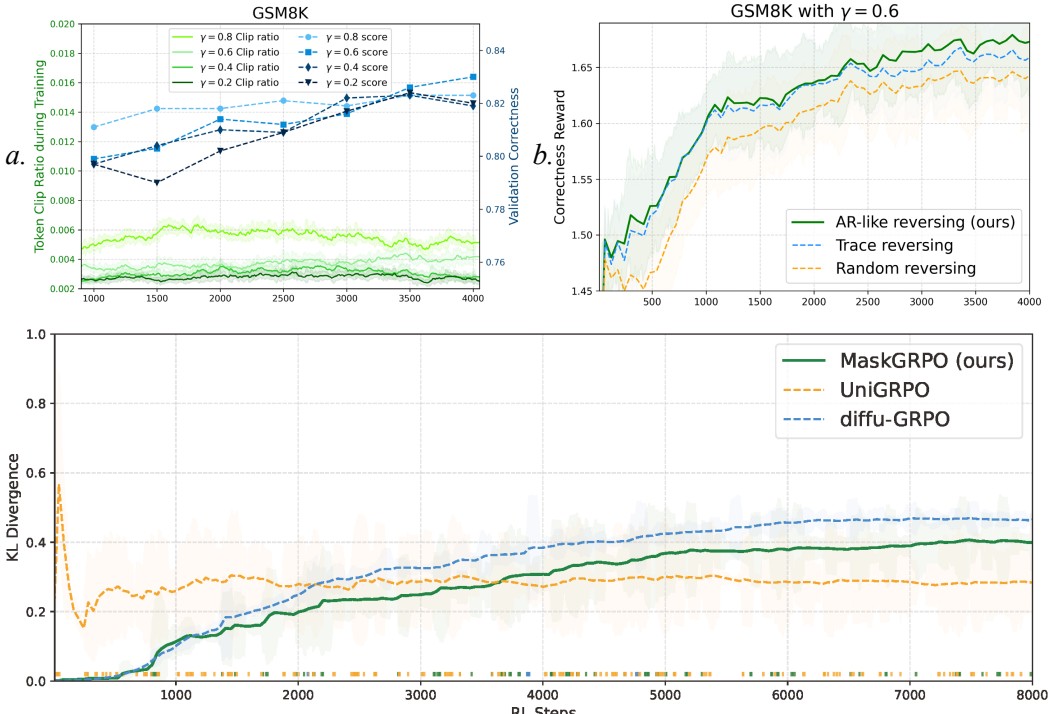

Figure 5: **Ablative results.** *a*: ablation on timestep truncation in language tasks. *b*: ablation on reverse methods in language tasks. Bottom: KL divergence during RL training under different reverse strategies. See text for detailed explanation.

clarity of visualization, we discard divergence values above 1.5 prior before smoothing and plot these filtered points as markers above the x-axis to indicate their frequency.

In math-reasoning tasks, occasional large divergence values are expected: they signal that the policy is exploring predictions that differ substantially from the reference model. However, frequent spikes can introduce instability and slow convergence. UniGRPO exhibits high divergence from the outset and continues to generate sharp fluctuations throughout training, indicating unstable updates. By contrast, *diffu*-GRPO, owing to its deterministic masking of all response regions during reversal, largely eliminates such spikes but also restricts exploration. MaskGRPO achieves a more favorable balance: it maintains stable training dynamics comparable to *diffu*-GRPO while preserving a healthy level of divergence that supports effective exploration. Please note that, while KL divergence provides useful diagnostic insight, it should not be treated as a standalone measure of training quality. A complete assessment requires considering performance trends alongside final task outcomes.

**Rollout comparison**   We investigate how the proposed emerge sampler improves generation quality under reinforcement learning. As an initial step, we substitute the vanilla MaskGIT style sampler with our method and evaluate the performance on MMaDA using GenEval. Before RL, our method produces samples with better textures, but the GenEval score is worse than that of the vanilla method (0.51 vs. 0.56). This discrepancy arises because some of our outputs exhibit unstable or deformed object boundaries, which negatively affect detector-based metrics.

After RL training, however, these instabilities are largely eliminated. The emerge sampler not only facilitates broader exploration during policy optimization, but also guides the model toward higher-quality local optima that would otherwise be inaccessible to MaskGIT sampling. Consequently, our method achieves a higher GenEval score (0.81 vs. 0.77), while also producing more stable and expressive generations. See 6. This progression highlights a key advantage: although the emerge sampler may underperform at the pre-RL stage, its enhanced exploration dynamics ultimately lead to stronger convergence and superior sample quality compared to the vanilla baseline.

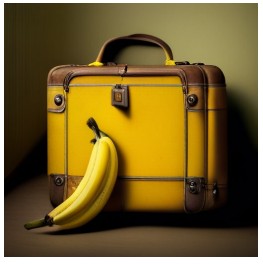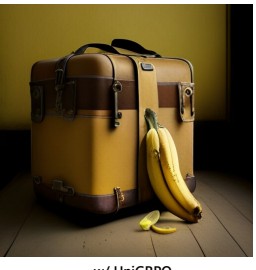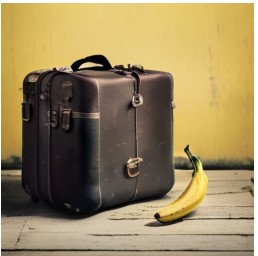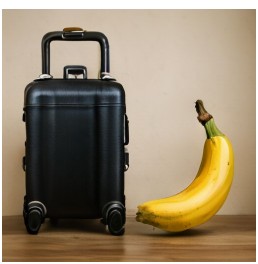

| Before GRPO | w/ UniGRPO | w/ MaskGRPO | w/ SFT + MaskGRPO |

Figure 6: **Visualization on GenEval.** Prompt: *a photo of a suitcase left of a banana.*

**Ablation on design choices.** We evaluate the impact of our reverse and sampling choices within MaskGRPO. The results, summarized in Tables 4 and 5, demonstrate the effectiveness of each component.

For GSM8K (256 completion length), the transition to *AR-like reversing* provides the largest single jump in performance. This confirms that utilizing the autoregressive bias is critical for importance estimation, aligning with the divergence analysis.

In the visual domain, the *emerge sampler* amplifies the gains to more than 20 points. Finally, the algorithmic modifications including ratio truncation act as a universal stabilizer, consistently improving performance across both modalities, confirming that MaskGRPO effectively consolidates these RL improvements for DDMs.

Table 4: Ablation on math reasoning.

| Design | GSM8K | Δ |
|---|---|---|
| Baseline (*diffu*-GRPO) | 79.8 | +3.1 |
| + Managed Randomness | 80.4 | +3.7 |
| + AR-like Rev. | 83.5 | +6.8 |
| + Truncation, *etc.* | **84.7** | **+8.0** |

Table 5: Ablation on image generation.

| Design | GenEval | Δ |
|---|---|---|
| Baseline (UniGRPO) | 0.63 | +0.07 |
| + Truncation, *etc.* | 0.75 | +0.19 |
| + Emerge Sampler | **0.81** | **+0.25** |

## 5 CONCLUSION

In this work, we introduced MaskGRPO, a modality-aware extension of Group Relative Policy Optimization for discrete diffusion models. Recalling rollout sampling and likelihood estimation, we developed tailored strategies for language and vision generation: fading-out masking for text and probabilistic decoding for images. Our experiments demonstrate that these design choices substantially improve reasoning accuracy, text-image alignment, and sample diversity. These results highlight the importance of modality-specific samplers and estimators for effective policy optimization, and pave the way for unified reinforcement learning approaches across multimodal discrete diffusion.

## 6 ACKNOWLEDGMENTS

This work was supported by National Natural Science Foundation of China under Grant 62225208, 62521007 and CAS Project for Young Scientists in Basic Research under Grant No.YSBR-117.

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

## A  ETHICAL STATEMENTS

We have provided key algorithms in the main text. Further implementation details are available in the source code. In the process of drafting this paper, Large Language Models (LLMs) were solely utilized for **grammatical checking**. No LLM was involved in core academic work such as conceptualization, literature review, data analysis, or argument construction of this study.

As the human authors of this paper, we bear full and sole responsibility for the paper's content, including the accuracy of research data, validity of academic arguments, integrity of research methods, and compliance with academic ethics.

## B  RELATED WORK

**Discrete Visual Diffusion Models**   Discrete diffusion, or its core idea of predicting on multiple discrete targets, has been applied to visual generation with prior endeavors like MaskGIT Chang et al. (2022), where a low-confidence re-mask sampler with Gumbel noise is used for inference to enhance sample quality. However, this can limit output diversity and makes reliable likelihood estimation challenging (Zheng et al., 2025; Ma et al., 2025a), creating a bottleneck for online RL.

**Text Diffusion Models**   Discrete diffusion models have emerged as powerful alternative (Sahoo et al., 2024; Shi et al., 2024; Nie et al., 2025) to autoregressive paradigms for language generation. Recent works Zhu et al. (2025); Gong et al. (2025) have demonstrated competitive performance in reasoning tasks. To obtain optimal results, while training in fully random noised corpses, current state-of-the-art performances are usually obtained via semi-autoregressive decoding (Arriola et al., 2025; Nie et al., 2025). While this *inconsistency* leaves space for more sophisticated training design, it also partially demonstrates the causal nature of language modality.

**Group Relative Policy Optimization**   GRPO and related reward-based optimization techniques have shown success in reinforcement learning for text generation and reasoning in autoregressive LLMs (Shao et al., 2024). As continuous diffusion has been integrated into the unified model's framework (Deng et al., 2025; Zhou et al., 2025), Liu et al. (2025) also adapt this RL paradigm in recent works via designed SDE solver. However, GRPO's application is fundamentally limited in discrete diffusion models, as it renders intractability on computing importance sampling weight, due to the lack of a factorized likelihood (Zhu et al., 2025).

## C  DISCUSSING THE APPROXIMATION

### C.1  ELBO ESTIMATION

The intractable log-likelihood $\log \pi_\theta(y|x)$ in DDMs is often approximated from its ELBO:

$$\mathcal{B}_\pi(y|x) \triangleq \mathbb{E}_{t \sim \mathcal{U}(0,1)} \mathbb{E}_{y^t \sim q(y^t|y,t)} \ell_\pi(y^t, t, y|x) \leq \log \pi(y|x). \tag{12}$$

And LLaDA-1.5 provided an estimation with proved low variance as

$$\hat{\mathcal{B}}_\pi(y|x) = \frac{1}{\nu} \sum_{j=1}^{\nu} \frac{1}{t_j} \sum_{k=1}^{|y|} \delta(y_k^{t_j}, \mathbf{m}) \log \pi_\theta(y_k^{t_j}, y|x) \approx \log \pi(y|x), \tag{13}$$

where $t_j = j/\nu$ is a discretization of $t$, and $y^{t_j}$ is sampled as $y^{t_j} \sim q(y^{t_j}; y, t_j)$. In practice, the time-weighted term is absorbed in to $\ell_\pi$ as an average on masked tokens, with the simplified form written as:

$$\hat{\mathcal{B}}_\pi(y|x) = \frac{1}{\nu} \sum_{j=1}^{\nu} \ell_\pi(y^{t_j}, y|x), \tag{14}$$

### C.2  IMPORTANCE ESTIMATION

We discuss the importance estimation in Eq. 8. Note that we do not intend to establish a strict proof for this, but kindly discuss the viable implementation which is shared among current RL works. First, let $o$'s subscript index $k$ temporarily represent its $k$-th element, we recall the loss expression as

$$\ell_\pi(o^t, o|\mathbf{c}) \triangleq \sum_{k=1}^{|o|} \delta(o_k^t, \mathbf{m}) \log \pi(o_k|o^t, \mathbf{c}). \tag{15}$$

Given partially masked completion $o^t$, the above term describes deviation between model's prediction $\pi$ on $o^t$'s masked positions. We also use $\dot{o}^t = o^t - o^{t+\delta t}$ to denote the tokens that are unmasked at timestep $t$, with $\delta t = \frac{|\dot{o}^t|}{|o|}$. For clarity, let $x = (c, o^{t+\delta t})$, $y = \dot{o}^t$. Using the low-variance estimation

in Eq. 14, we discuss Eq. 8 as follows:

$$\log \pi_1(y|x) - \log \pi_2(y|x) \approx \hat{\mathcal{B}}_{\pi_1}(y|x) - \hat{\mathcal{B}}_{\pi_2}(y|x)$$
$$= \frac{1}{\nu} \sum_{j=1}^{\nu} \left[ \ell_{\pi_1}(y^{t_j}, y|x) - \ell_{\pi_2}(y^{t_j}, y|x) \right] \quad (16)$$

Considering the small incremental of $y$ comparing to $x$, and the computational resource allocated to the inner-loop $\nu$ is limited, we can make further approximation by calculating on step $t_j = 1$, where all tokens in $\dot{o}^t$ are pending:

$$\frac{1}{\nu} \sum_{j=1}^{\nu} \left[ \ell_{\pi_1}(y^{t_j}, y|x) - \ell_{\pi_2}(y^{t_j}, y|x) \right] \approx \ell_{\pi_1}(\mathbf{m}^{|y|}, y|x) - \ell_{\pi_2}(\mathbf{m}^{|y|}, y|x)$$
$$= \ell_{\pi_1}(\dot{o}^t, o|\mathbf{c}, o^{t+\delta t}) - \ell_{\pi_2}(\dot{o}^t, o|\mathbf{c}, o^{t+\delta t})$$
$$= \ell_{\pi_1}(\dot{o}^t + o^{t+\delta t}, o|\mathbf{c}) - \ell_{\pi_2}(\dot{o}^t + o^{t+\delta t}, o|\mathbf{c})$$
$$= \ell_{\pi_1}(o^t, o|\mathbf{c}) - \ell_{\pi_2}(o^t, o|\mathbf{c}) \quad (17)$$

■

## D    ADDITIONAL MATERIAL

### D.1    DIVERSITY IN GENERATION TRAJECTORIES

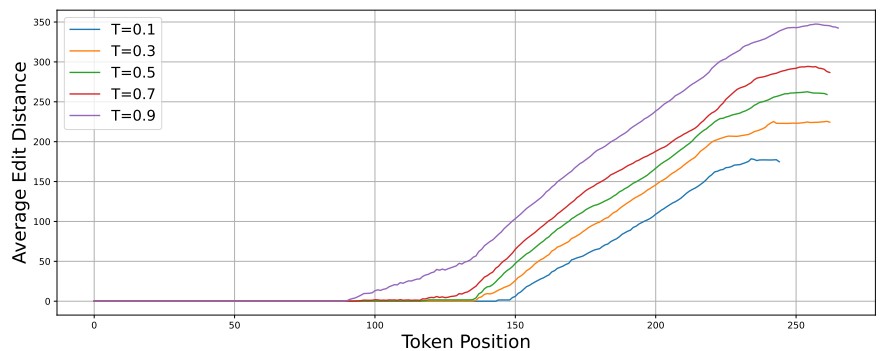

Figure 7: A visualization of diversity in trajectories.

We provide an empirical support for the observation that text tokens exhibit increasing trajectory diversity as generation progresses. For a fixed prompt, we generate $N = 16$ independent continuations across different sampling temperatures $(T)$, which directly control the output entropy. The generation parameters are fixed at `gen_len/step/block_len = 256/128/16`. The curves in Figure 7 plot the Average Pairwise Levenshtein Edit Distance among partial generations up to token positions. The consistently positive slope of all curves confirms that the divergence between generation trajectories grows significantly as token position extends. Furthermore, the sensitivity to $T$ shows that higher $T$ accelerates this divergence, resulting in a steeper curve, while lower temperatures (e.g., $T = 0.1$) maintain higher stability and determinism in the early stages of generation, validating the premise for focusing on later tokens in LLaDA's rollouts. During GRPO training, the temperature is set to $T = 0.9$ for language tasks (the default value of GRPO trainer, same as other methods) throughout the training process.

### D.2    IMAGE EDITING EXAMPLES

We conducted a preliminary study on image editing, which is a key capability for modern generative models and remains largely underexplored for discrete approaches. We chose the PIE-Bench Ju et al. (2024) test suite for evaluation. Representative examples of the results are included in Fig, 8.

Target: a [fox] is shown [smiling] in a [high polygonal style]

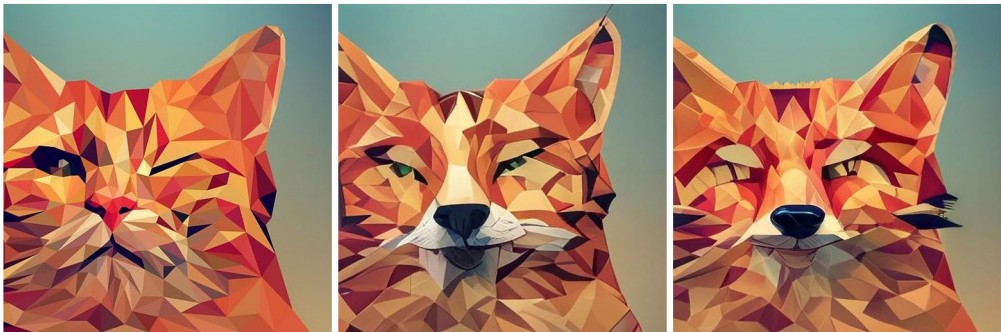

Target: a [white] plate with grilled [salmon] [and mushroom] on it

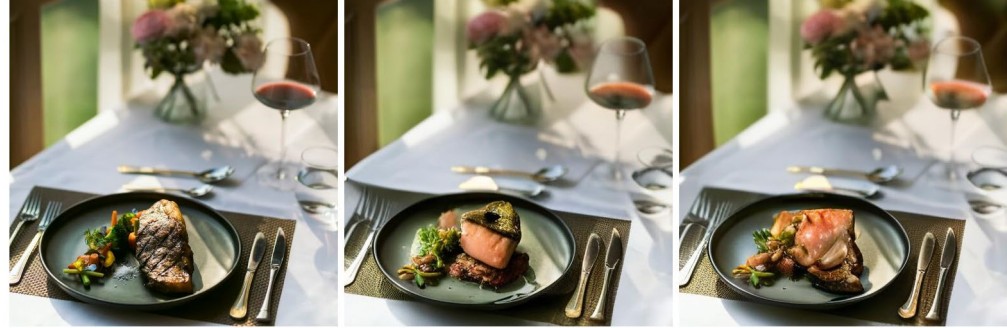

Target: a [parrot] on a [flower] in the garden

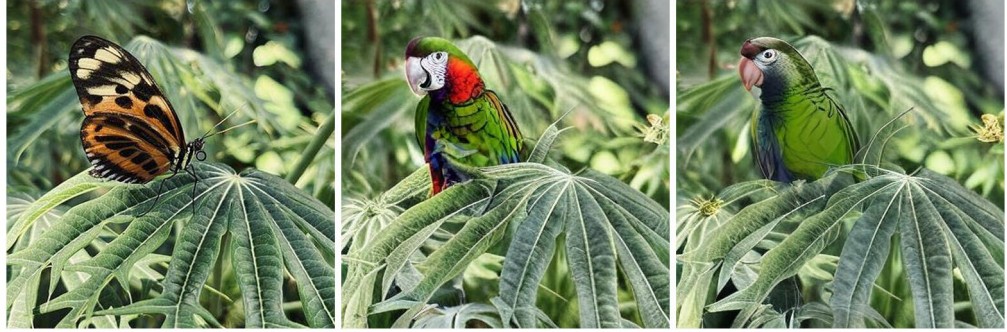

Target: a [realism style] [painting of] bird standing on tree branch

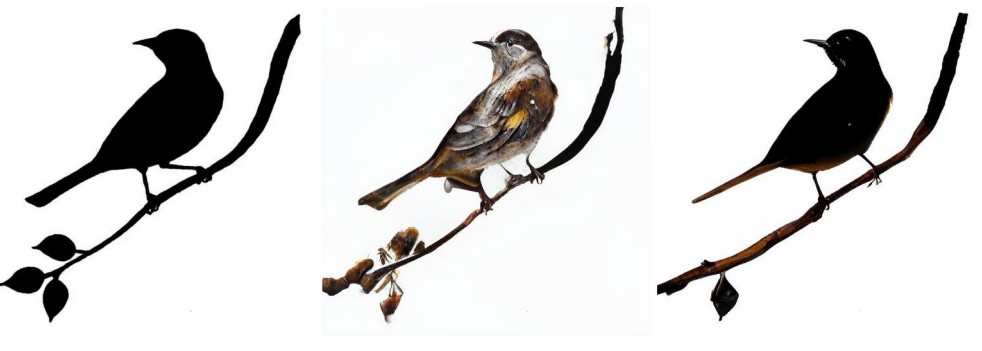

Original Image     **Emerge method (ours)**     MaskGIT method

Figure 8: A visualization of image editing examples.

The image editing process relies on a reverse masking strategy to selectively keep or remove parts of the image tokens $o$ before regeneration. We investigated two distinct reverse masking methods: **(1) Baseline (Random Masking):** This method involves re-masking 70% of the total image tokens, corresponding to $o^{0.5}$ under the cosine scheduler. **(2) Score-based Masking:** To preserve the image structure relevant to the original content, we input the original image and the modified prompt into the model. We extract the predicted logits of the original image's corresponding tokens as "scores," which estimate the model's confidence in preserving the original content. We then mask the 70% of tokens associated with the least scores, thereby attempting to keep the most confident structural elements unmasked.

The re-masked image sequence is subsequently fed into the model for the completion steps. We use **9** additional steps to generate the edited image. We compared the standard MaskGIT method with the Emerge sampler under this editing setting. The quantitative results are summarized in Table 6.

Table 6: Quantitative results for Image Editing on PIE-Bench.

| Masking Method | Sampler | Structure Distance ↓ | LPIPS ↓ | SSIM ↑ |
|---|---|---|---|---|
| Random | MaskGIT | 141.90 | 410.13 | 50.78 |
| Random | **Emerge** | **103.68** | **270.07** | **60.52** |
| Score-based | MaskGIT | 23.55 | 90.91 | 77.44 |
| Score-based | **Emerge** | **21.90** | **87.40** | **78.07** |

The results demonstrate that the Emerge sampler consistently produces more coherent and higher-fidelity edits, as evidenced by superior metrics (lower Structure Distance, lower LPIPS, and higher SSIM) across both masking strategies. Notably, the combination of score-based masking with the Emerge sampler yields the best performance.

That said, we note that strong image editing performance typically requires large-scale supervised training, which MMaDA was not trained on. Consequently, these results serve as early evidence of the framework's potential in image manipulation.

# E  IMPLEMENTATION DETAILS

## E.1  TRAINING

For language tasks, Following the practice of *diffu*-GRPO, we conduct with a similar learning rate of $3e^{-6}$, rollouts per prompt $G = 6$, iteration $\mu = 12$, and a global batch size of 96 (bs = 6 on $8\times$A100 GPU, with gradient accumulation $n = 2$). The rollout is sampled with a block length of 16, and 2 tokens per step. Let global steps be the update steps divided by iterations, we train max = 500 global steps on all tasks, which takes up to 25% fewer training steps than that of *diffu*-GRPO.

For GenEval task, we use rollouts per prompt $G = 8$ for exploration, iteration $\mu = 8$, accumulation $n = 4$, and a global batch size of 256. Each rollout is sampled with our emerge sampler, using 12 steps at CFG= 3.5 with a cosine scheduler. The RL training takes 1500 global steps.

## E.2  EVALUATION

For language tasks, we evaluate all tasks with 0-shot prompting. We use a block length of 16 and decodes 2 tokens per step for math tasks, and the MBPP protocol is specified in the following paragraph. All performances are reported using the pass@1 metric. For image generation, the sampler decodes an visual sequence of 1024 tokens (which represents an image with resolution $512 \times 512$) with 32 steps, and is equipped with classifier-free guidance at 3.5, consistent with the original MMaDA configuration.

**MBPP Evaluation Protocol.**  We specify the standardized protocol used for evaluating models on the Mostly Basic Python Problems (MBPP) benchmark. We clarify this protocol to address the significant variance in results reported in the literature, which stems from inconsistent settings for

generation parameters (`gen_len/step/block_len`), different prompt designs, and distinct data subsets (e.g., `sanitized-mbpp.json`). Such variations impede direct model comparisons.

Following the evaluation setup of LLaDA-8B-Instruct, we specify our standard as follows: The test set consists of the first 500 samples (1-500) from the `mbpp.jsonl` file in the official dataset. The evaluation is conducted in a zero-shot setting, using the same prompt format as the `lm-eval` library. The generation parameters are fixed at `gen_len/step/block_len = 256/256/32`. Performance is reported using the pass@1 metric, which measures the percentage of test cases passed on the first attempt.

### E.3 DATA USAGE

**Language tasks.** We use the standard training sets for GSM8K and MATH500. For MBPP, we follow DiffuCoder and use Acecode-87K, an open-source code dataset. Notably, *diffu*-GRPO reported using KodCodeLight-RL-10K, which refers to multiple traverse over the dataset, given its reported 7500 steps. Besides, as the corresponding implementation is missing from its codebase, we re-implement it and discovered limited effectiveness. We report its best performance on MBPP with our re-implementation on Acecode-87K.

**Image generation tasks.** For the results reported in 2, we use filtered 15K GenEval-style prompts (provided in the repository). For general prompt following ability, we followed X-Omni by randomly sampling 90K prompts from `midjourney-prompts`, a dataset of real user instructions, and augment it with 60K compositional GenEval-style prompts from Blip3-o (guaranteed that there is no overlap with the benchmarks).

Additionally, we utilize instruction tuning data from Blip3-o for SFT. This procedure is optional, and we have denoted the corresponding results with explicit SFT mark in the tables. As observed, MMaDA's prompt following ability can be improved with such extremely clean supervision signals, and the generated images have a more accurate demonstration of spatial relationship, and clearer boundary among objects, compared to soley RL results. We train on this dataset for $\sim 1000$ steps with global batch size 128, and a learning rate of $3e^{-6}$.

### E.4 REWARD DESIGN

**Language tasks.** For GSM8K and MATH500, the reward consists of two components:

- Correctness reward: returns 2 for an extracted and correct final answer, and 0 otherwise.
- Format reward: returns 0.5 if the reasoning process is properly enclosed in `<reasoning>*</reasoning>`, and 0 otherwise.

For MBPP, we adopt DiffuCoder's scheme, combining correctness and format rewards. The format reward ensures completions are wrapped in `''' * '''`, while the correctness reward tests generated code against predefined test cases.

**Image generation tasks.** For image generation tasks, the reward can be calculated with:

- **UnifiedReward**, evaluates image–prompt alignment, divided by 5 to $[0, 1]$.
- **HPSv3**, assess visual quality and text–image alignment, the score is divided by 5 to an approximate range $[0, 2]$.
- **CLIP Score**, measures similarity between encoded text and image features, ranging from $[0, 1]$ (typically 0.2–0.4). We retain this metric to mitigate reward hacking.

We recommend using the sum of these three components. In the updated version, we also support using single GenEval score as the default reward, the results are shown in 2.

### E.5 CLARIFICATION

We select LLaDA-8B-Instruct for language tasks and MMaDA-8B-Base for image generation tasks. Both models share a similar architecture and are initialized from LLaDA-8B-Base. While this choice

does not affect our claims on multimodal reinforcement learning, we clarify our rationale: although MMaDA released a MixCoT checkpoint, its performance on math and coding tasks is severely limited. With reasoning enabled, it achieves only 48% accuracy on GSM8K, about 30% lower than LLaDA-8B-Instruct with same sampling parameters. We attribute this to potentially insufficient training or a suboptimal recipe leading to catastrophic forgetting. Although MaskGRPO applied to MMaDA-8B-MixCoT yields an improvement of over 6%, the results are not comparable since prior works consistently use LLaDA as the baseline.

# F  MORE RESULTS

## F.1  VISUALIZATION OF REVERSE METHODS

We provide a visualization of trace reversing Wang et al. (2025b) and AR-like reversing methods in Fig. 9 with block length 16. AR-like method balances the autoregressive bias and randomness, while trace method reverses the decoding path with rigid paths. The strict trace-based method may also suffer from coarse approximations when applied to highly parallel decodings, *e.g.,* visual generation scenarios, where hundreds of tokens will be updated within only a few steps.

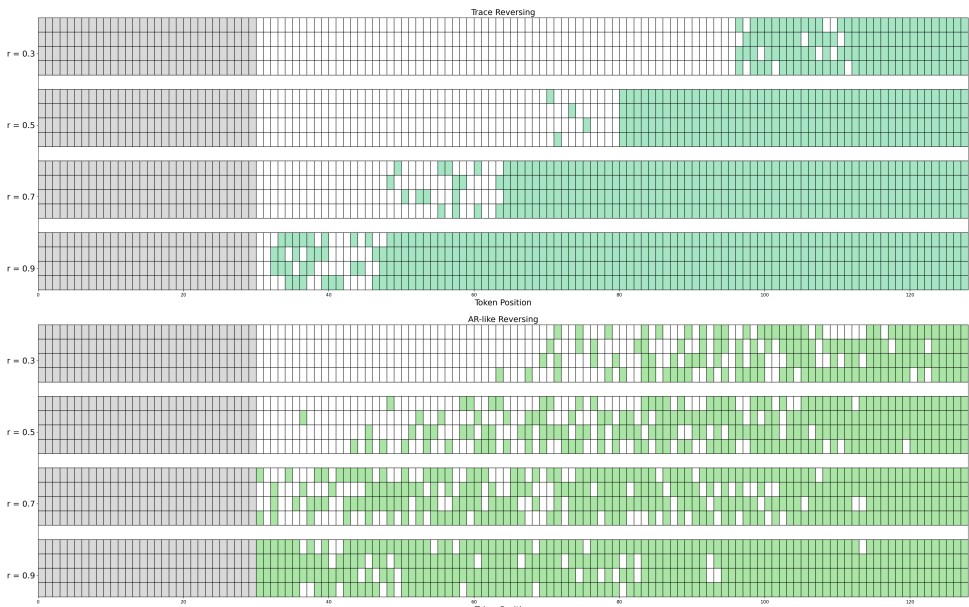

Figure 9: A comparison of trace reversing (above) and AR-like reversing (below, ours).

## F.2  IMAGE GENERATION

We report the complete evaluation results on DPG-Bench in Tab. 7. We also provide more generated portrait samples in Fig. 10.

Table 7: **Evaluation on DPG-Bench.** *SFT* indicates that we tune the base model with BLIP3-o dataset (distilled from GPT-4o) for clean supervision data before RL training.

| Model | DPG-Bench↑ | | | | | |
|---|---|---|---|---|---|---|
| | **Global** | **Entity** | **Attribute** | **Relation** | **Other** | **Overall** |
| SDXL Podell et al. (2023) | 83.27 | 82.43 | 80.91 | 86.76 | 80.41 | 74.65 |
| Hunyuan-DiT Li et al. (2024) | 84.59 | 80.59 | 88.01 | 74.36 | 86.41 | 78.87 |
| Emu3-Gen Wang et al. (2024) | 85.21 | 86.68 | 86.84 | 90.22 | 83.15 | 80.60 |
| SD3-Medium Esser et al. (2024) | 87.90 | 91.01 | 88.83 | 80.70 | 88.68 | 84.08 |
| MMaDA Yang et al. (2025) | 77.52 | 77.52 | 77.67 | 81.56 | 63.13 | 70.51 |
| w/ MaskGRPO (ours) | 80.92 | 79.52 | 85.41 | 83.05 | 69.12 | 75.81 |
| w/ SFT+MaskGRPO (ours) | 85.96 | 84.05 | 89.76 | 83.43 | 79.88 | 81.76 |

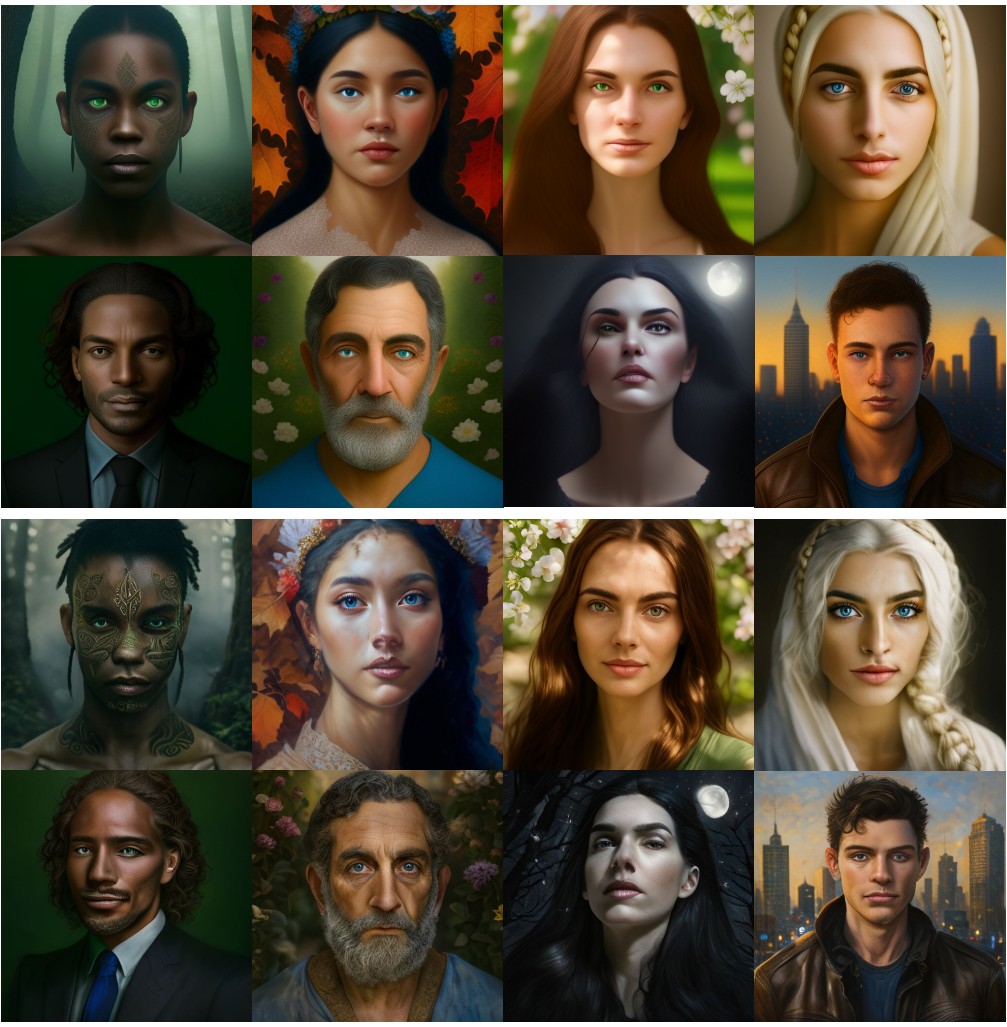

Figure 10: More generated portrait samples with identical prompt and sampling parameters (CFG=3.5, 64 step). Above: sampled by default MaskGIT-style sampler before RL training. Below: sampled by our emerge sampler after RL training. Prompts are available in the supplementary material.

F.3 MATH REASONING

---

**Generation Sequence Length:** 256
**Question:** There are 29 pupils in a class. The teacher has 9 coupons; each coupon can be redeemed for 100 bottles of apple juice. The teacher gives each student 2 bottles of apple juice to drink for lunch. After redeeming all her coupons and giving each student their apple juice lunch how many bottles of apple juice does the teacher have for herself?

<reasoning> First, we need to determine the total number of bottles of apple juice the teacher can redeem with her coupons. Each coupon can be redeemed for 100 bottles, and the teacher has 9 coupons. Therefore, the total number of bottles the teacher can is:

$$9 \times 100 = 900$$

Next, we need to calculate the total number of bottles of apple juice given to the students. There are 29 pupils in the class, and each pupil receives 2 bottles. Therefore, the total number of bottles given to the students is:
$$29 \times 2 = 58$$

Now, we subtract the number of bottles given to the students from the total number of bottles the teacher can redeem to find out how many bottles the teacher has for herself:
$$900 - 58 = 842$$

Thus, the teacher has 842 bottles of apple juice for herself. < /reasoning> 842

---

**Generation Sequence Length:** 512
**Question:** A watermelon farm produced 120 baby watermelons. Ashlyn went to the farm after two months and found out that 30% of the watermelons were ready for harvest so she took them home. When she came back two weeks later 3/4 of the remaining melons were ready so she harvested them. How many melons were not ready to be harvested after the two weeks?

<reasoning> To determine how many watermelons were not ready to be harvested after two weeks, we need to follow these steps:
1. Calculate the number of watermelons ready for harvest after two months.
2. Subtract the number of watermelons harvested after two months from the total number of watermelons to find the remaining watermelons.
3. Calculate the number of watermelons ready for harvest two weeks later.
4. Subtract the number of watermelons harvested two weeks later from the remaining watermelons to find the number of watermelons not ready to be harvested.
First, let's calculate the number of watermelons ready for harvest after two months:

$$30\% \text{ of } 120 = 0.30 \times 120 = 36$$

So, 36 watermelons were ready for harvest after two months. Next, we subtract the number of watermelons harvested after two months from the total number of watermelons to find the remaining watermelons:
$$120 - 36 = 84$$

So, there are 84 watermelons remaining after two months. Now, let's determine how many watermelons were ready for harvest two weeks later:

$$\frac{3}{4} \text{ of } 84 = \frac{3}{4} \times 84 = 63$$

So, 63 watermelons were ready for harvest two weeks later. Finally, we subtract the number of watermelons harvested two weeks later from the remaining watermelons to find the number of watermelons not ready to be harvested:

$$84 - 63 = 21$$

So, 21 watermelons were not ready to be harvested after two weeks. < /reasoning> 21

---

## F.4 CODE GENERATION

**Generation Sequence Length:** 256
**Question:** You are an expert Python programmer and here is your task: Write a function to find the vertex of a parabola.

**Response:** To find the vertex of a parabola given by the equation

$$y = ax^2 + bx + c$$

you can use the following formulas:
1. The x-coordinate of the vertex is given by

$$x = -\frac{b}{2a}$$

2. The y-coordinate of the vertex is given by

$$y = c - \frac{b^2}{4a}$$

Here's a Python function that implements these formulas:

```
def parabola_vertex(a, b, c):
    x_vertex = -b / (2 * a)
    y_vertex = c - (b**2) / (4 * a)
    return (x_vertex, y_vertex)
```

**Test cases:**
"assert parabola_vertex(5,3,2)==(-0.3, 1.55)", $True$

"assert parabola_vertex(9,8,4)==(-0.4444444444444444, 2.2222222222222223)", $True$

"assert parabola_vertex(2,4,6)==(-1.0, 4.0)", $True$

