# OpenReview forum: "Consolidating Reinforcement Learning for Multimodal Discrete Diffusion Models"
_ICLR.cc/2026/Conference — ICLR 2026 Poster_

### Official Review · Reviewer_pNLG · 2025-10-17

**Soundness:** 3
**Presentation:** 3
**Contribution:** 3
**Rating:** 8
**Confidence:** 2

**Summary:**

The authors propose maskGRPO, a GRPO-style RL method for discrete diffusion models developed and shown to work on multimodal DDMs, both text and images. To achieve this, they propose a tractable importance estimator, and an approximated surrogate for importance weighting and KL in GRPO for DDMs. The authors claim more stable and efficient updates while showing improved performance on a set of both visual and language tasks.

**Strengths:**

The paper is generally well written and easy to follow. The results are compelling enough and it's nice to see GRPO working on image DDMs. The formulations of the importance estimator for DDMs is also useful.

**Weaknesses:**

A cornerstone of the narrative of the paper seems to revolve around compute/sample efficiency, but I did not see a detailed tokens-to-gain or wall-clock comparison against the strongest off-policy GRPO / surrogate-policy approaches, or a theoretical note explaining why the method is expected to work more efficiently in practice.

**Questions:**

Can you provide either empirical or theoretical grounding in support of markGRPO being more efficient than similar methods?

---

> ### Author Response · Authors · 2025-11-24
>
> Thank you for the encouraging feedback. We appreciate your comments on the clarity of the writing, the usefulness of our importance estimator, and the effectiveness of applying GRPO to image DDMs. Below we provide point-by-point responses.
> ___
>
> **Weakness: ... a detailed tokens-to-gain or wall-clock comparison ... or a theoretical note explaining why the method is expected to work more efficiently in practice.** + **Question: Can you provide empirical or theoretical grounding in support of markGRPO being more efficient than similar methods?**
>
> **Response:**
> Thank you for raising this important point.
> As the current research for reinforcing DDMs is still at early stage, we follow the first available baseline method, Diffu-GRPO closely by using the same TRL framework, optimizer, and hyperparameters. The main differences lie in:
> 1. Deterministic randomness control, which does not incur additional cost.
> 2. AR-like reversing for importance sampling, which is actually more efficient than Diffu-GRPO’s fixed-ratio reversing.
>
> Using the identical settings (μ = 12, rollout per prompt G = 6) and training on 8×A100 GPUs, we replicated experiments:
>
> | Method     | Best Step | GSM8K Score | Improvement | Wall-clock |
> |------------|-----------|-------|-------------|-------------|
> | Diffu-GRPO | 6600      | 79.8  | +3.1        | ~20 hours   |
> | UniGRPO    | 6000      | 81.1  | +4.4        | ~19 hours   |
> | MaskGRPO   | 4500      | 84.7  | +8.0        | ~14 hours   |
>
> On language tasks, MaskGRPO achieves higher accuracy, larger gains, and **~30%** faster training. We also revise the main paper (marked with teal) with detailed discussion on dynamics of KL divergence during RL, ablation on each proposed component, and additional experiments.
>
> As for visual generation, MaskGRPO is the first viable RL framework for reinforcing DDMs like MMaDA, and it surpasses the baseline (UniGRPO, which has no training protocol specified) by up to **20%**.
>
> The efficiency of MaskGRPO arises from the synergistic design across modalities and algorithmic components:
>
> * AR-like reverse improves importance sampling quality and stabilizes gradients.
> * Emerge sampler leverages multimodal token dynamics for more reliable rollouts.
> * Controlled randomness reduces variance and improves sample efficiency.
> * DDM-specific theoretical insights guide the construction of our importance estimator, ensuring gradient updates reflect meaningful token fluctuations.
>
> Together, these elements create a well-engineered, modality-aware RL framework that is both more effective and more computation-efficient than prior approaches.
> ___
>
> We hope the above clarification addresses your concerns and is helpful for your final assessment. Thank you again for the thoughtful feedback.

---

> > ### Comment · Reviewer_pNLG · 2025-11-25
> >
> > Thank you for addressing my concern, I maintain my score.

---

> > > ### Author Response · Authors · 2025-11-25
> > >
> > > Thanks for your thoughtful feedback and for taking the time to review our work. We’re glad to hear that the our response helped address your concerns!

---

### Official Review · Reviewer_Extk · 2025-10-30

**Soundness:** 3
**Presentation:** 3
**Contribution:** 3
**Rating:** 8
**Confidence:** 3

**Summary:**

The paper proposes MaskGRPO, an RL framework extending GRPO to multimodal DDMs. It introduces modality specific rollout strategies with AR-like reversing for text and probabilistic emerging for vision, along with a low-variance importance estimator for stable and scalable optimization across discrete diffusion. Experiments show consistent gains in reasoning, coding, and visual generation.

**Strengths:**

1. The problem is clearly defined — the paper directly targets the limitation of static embeddings in collaborative code completion by proposing a dynamic, incremental embedding approach.

2. The overall framework is well-structured and conceptually coherent, with a clear system design.

3. The method improves both language and vision generation, demonstrating strong empirical utility.

**Weaknesses:**

The paper lacks discussion of training cost or comparison of resource consumption with baseline methods.

**Questions:**

1. How is the mask ratio in the Rev(·, t) operator scheduled? Is it fixed or dynamically adjusted during RL training?

2. How does the training cost of MaskGRPO compare with other diffusion-RL methods such as Diffu-GRPO?

3. The paper introduces modality-specific rollout strategies, but it is unclear how sensitive the performance gains are to these design choices. Could the authors provide ablation or analysis showing how each component (AR-like reversing, emerging sampler) contributes to the overall gains?

---

> ### Author Response · Authors · 2025-11-24
>
> Thank you for the positive feedback. We appreciate your recognition of the clear problem formulation, coherent system design, and strong empirical results across language and vision tasks. Below we provide point-by-point responses.
>
> ___
>
> **Q1: How is the mask ratio in the Rev(·, t) operator scheduled? Is it fixed or dynamic?**
>
> **A1:**
> Great question! The Rev(·, t) operator controls mask ratios by setting a target ratio and then using randomized thresholds to determine which tokens to mask, as described in Algorithms 1 and 2.
>
> For a given response, we use μ reverse stages, with ratios uniformly spanning from γ to 1. For ratio $r$, we treat this value as a threshold and generate a random matrix with values in [0,1]; elements below the threshold are masked. Note that, while this procedure is fully automated, the actually reversed tokens' ratio may not equals to $r$ due to randomness.
>
> Figure 2 can be regarded as a simulation that happens at [**algorithm 5, step 9**] during GRPO, where $r=0.6$, and 8 rollouts are AR-like reversed on 8 GPUs (supposing that train_batch_size_per_devise=1).
>
> We keep the schedule (algorithm 5) throughout the process, ensuring each response is evaluated over a spectrum of mask ratios and contributes effectively to the gradient.
>
> ___
>
> **Q2 (combined with weakness): The paper lacks discussion of training cost or comparison of resource consumption with baseline methods. How does MaskGRPO’s training cost compare with baselines such as Diffu-GRPO?**
>
> **A2:**
> Thank you for the question. Here is a direct comparison under equivalent training setups.
>
> We follow Diffu-GRPO closely by using the same TRL framework, optimizer, and hyperparameters. The main differences lie in:
> 1. Deterministic randomness control, which does not incur additional cost.
> 2. AR-like reversing for importance sampling, which is actually more efficient than Diffu-GRPO’s fixed-ratio reversing.
>
> Using the identical settings (μ = 12, rollout per prompt G = 6) and training on 8×A100 GPUs, we replicated experiments:
>
> | Method     | Best Step | GSM8K Score | Improvement | Wall-clock |
> |------------|-----------|-------|-------------|-------------|
> | Diffu-GRPO | 6600      | 79.8  | +3.1        | ~20 hours   |
> | UniGRPO    | 6000      | 81.1  | +4.4        | ~19 hours   |
> | MaskGRPO   | 4500      | 84.7  | +8.0        | ~14 hours   |
>
> MaskGRPO achieves higher accuracy, larger gains, and ~30% faster training.
>
> We also revised the main paper (marked with teal) with detailed discussion on dynamics of KL divergence during RL, ablation on each proposed component, and additional experiments.
>
> ___
>
> **Q3: The paper introduces modality-specific rollout strategies, but it is unclear how sensitive the performance gains are to these design choices. Could the authors provide ablation or analysis showing how each component (AR-like reversing, emerging sampler) contributes to the overall gains?**
>
> **A3:**
> Thank you for highlighting this. As noted above, our improvements focus on re-designed reverse and sampling procedures within the existing GRPO framework. We provide detailed ablations (summarized below):
>
> | Design | GSM8K Score | Improvement |
> |---|---|---|
> | Baseline (Diffu-GRPO) | 79.8 | +3.1 |
> |    + Managed randomness | 80.4 | +3.7 |
> |    + AR-like reversing | 83.5 | +6.8 |
> |    + ratio truncation | 84.7 | +8.0 |
>
> | Design | GenEval Score | Improvement |
> |---|---|---|
> | Baseline (UniGRPO) | 0.63 | +0.07 |
> |   + emerge sampler  | 0.75 | +0.19 |
> |   + ratio truncation | 0.80 | +0.24 |
>
> Key findings:
>
> * On reasoning tasks: AR-like reversing consistently outperforms other reverse strategies. This aligns with the visualization in Fig. 5(b) of the main text.
> * On visual generation: The emerge sampler amplifies GRPO’s gain from roughly 7 points to ~20 points, confirming its importance.
>
> These results support our claim that MaskGRPO consolidates RL improvements for discrete diffusion.
>
> We have revised the main paper's **Section 4.3** to make our proposed designs more clear with detailed ablations.
> ___
>
>  We hope the clarifications above address your concerns and contribute to your final assessment. Thank you again for the thoughtful and constructive feedback.

---

> > ### Comment · Reviewer_Extk · 2025-11-25
> >
> > Thank you for your response. My concerns have been addressed, and I will keep my current score.

---

> > > ### Author Response · Authors · 2025-11-25
> > >
> > > Thank you for your thoughtful feedback and for taking the time to review our work. We’re glad to hear that the our response helped address your concerns!

---

### Official Review · Reviewer_q71V · 2025-11-01

**Soundness:** 2
**Presentation:** 2
**Contribution:** 2
**Rating:** 2
**Confidence:** 3

**Summary:**

The work proposes a method adapting GRPO for multimodal discrete diffusion models, by using modality specific sampling algorithms for rollouts (semi-autoregressive for text, and random probabilistic unmasking for images), as well as modality specific remasking strategies to use for updates in the GRPO objective (autoregressive-like remasking for text and standard random remasking for images). This is done along with limiting GRPO updates to trajectories in the time interval $(\gamma, 1)$. The modified GRPO method is used to fine-tune LLaDA for text reasoning tasks as well as MMaDA (a multimodal discrete diffusion model) for text-image alignment and aesthetic quality, where it is shown to outperform alternate RL methods.

**Strengths:**

1. A comprehensive set of baselines are evaluated for experiments.
2. The generated image samples in Figures 4, 7 are convincing in terms of improvement.
3. The ablations with truncation ratio $\gamma$ are helpful in establishing its impact (Figure 5).

**Weaknesses:**

1. The writing needs polishing and proofreading
    - There are numerous grammatical mistakes that interfere with the clarity of presentation
    - The citation format is incorrect (using `\citet` rather than `\citep` )
    - The Emerge sampler is described unclearly in text (lines 238-247)

2. The Emerge sampler (algorithm 4) doesn’t seem novel, and appears to be the same as the usual random unmasking from masked diffusion models (eg. without confidence based heuristics for unmasking). This is discussed in the original MDLM paper (Sahoo et al., 2024) as well as other early work on masked diffusion (eg. (Shi et al., 2024)). Despite this, the text appears to frame the sampler as a novel contribution (eg. line 244 ‘we refer to MDLM … and propose the … sampling strategy’ and line 247, ‘our sampler’). This should be clarified.

3. A number of claims are made which require more detail or justification:
    - The motivation for the autoregressive-like remasking invokes an observation that tokens with high entropy provide more informative signal for training, and that later tokens (in AR generation) tend to diverge more. This should be supported by some evidence from rollouts
    - Remasking with the $\mathrm{Rev}(\cdot, t)$ operator is asserted to be more stable and have low-variance (line 192, and also on line 189, the estimator is asserted to be “low-discrepancy”)  (I am assuming, compared to random remasking) - but this statement should be verified explicitly.
    - TraceRL is claimed to induce “biased estimation of sequence-level importance” - why is the method biased, compared to remasking with $\mathrm{Rev}(\cdot,t)$? This seems important since the introduction of a new remasking strategy is a core aspect of the method, and the straightforward thing appears to be reusing the partially masked completion obtained during rollouts.

4. It would be helpful to list the time taken to achieve the reward improvement (or some proxy), since the importance weight computation in this method appears to be more computationally intensive than the mean-field approximation for token level likelihood used in diffu-GRPO

I recommend for a reject, mainly due to point 2 above, which I view as critically important.

Jiaxin Shi, Kehang Han, Zhe Wang, Arnaud Doucet, and Michalis K Titsias. Simplified and
generalized masked diffusion for discrete data. arXiv preprint arXiv:2406.04329, 2024.

**Questions:**

1. For text, if tokens later on in generation (more towards the end of generation) are more useful for training, in what sense does the AR-like process assign them “higher attention”.
    - This appears to conflict with the truncation ratio being set at $\gamma > 0$  since it will exclude a group of tokens near the end of the (block-autoregressive) generation.

2. Is the masking rate multiplier ($\frac1t$ for linear) used in the likelihood computation at the token-level (for instance Equation 8)? The notation in Equation (2) implies it is.
    - If it is, is the masking rate adjusted for the alternate remasking strategies considered (namely autoregressive-like remasking)?

3. The mechanism behind why smaller truncation ratios lead to training failure (for image fine-tuning) is unclear. An explanation is given in terms of samples having stronger correlations between image patches, but its unclear to me why this results in collapse.

---

> ### Author Response · Authors · 2025-11-24
>
> Thank you for your recognition of the comprehensive baselines, the improved image samples, and the usefulness of the truncation-ratio ablations. Before addressing the remaining comments, we would like to respond to the following point first.
>
> **Question: The Emerge sampler (algorithm 4) doesn’t seem novel, and appears to be the same as the usual random unmasking from masked diffusion models... Despite this, the text appears to frame the sampler as a novel contribution...This should be clarified.**
>
> ____
>
> **Response:**  Thank you for highlighting this point. We apologize for any ambiguity in the original presentation. To clarify upfront: The emerge sampler is indeed inspired by MDLM [1], but it is **not a direct reuse** of prior samplers. It is a variant we developed through empirical investigation on generative DDMs like MMaDA, with modifications required for multimodal visual vocabularies and for consistent integration with classifier-free guidance (CFG) and DDM theory. As written in the abstract, we *delicately tailored the rollout method for visual sequences*, rather than claiming a fundamentally new paradigm. We have revised the paper text accordingly (marked in teal)
>
> Below we provide a more detailed clarification of the motivation and the specific implementation differences:
>
> **Motivation:**
>
> While examining current text-to-image discrete diffusion systems (e.g., MUSE, Meissonic, Show-o), we observed that they typically rely on MaskGIT-style sampling. MMaDA, while trained with modern DDM loss, still follows the same practice. This creates a mismatch between the training objective and the sampling procedure, leading to weak fine-grained details and overly uniform textures—effects that significantly degrade downstream RL performance.
>
> Our goal was therefore to adapt a sampler that
>
> * better accommodates large multimodal vocabularies,
> * integrates CFG correctly and efficiently, and
> * remains aligned with the underlying DDM formulation.
>
> To our knowledge, ours is the first discussion focused on quality-aware sampling for discrete visual diffusion. The submitted version summarized these motivations but did not enumerate the concrete implementation differences, which may have caused confusion. We now clarify these below.:
>
> **1. Logit handling.**
>
> MDLM applies a SUBS-parameterization followed by an exp() transformation due to its loss formulation, which introduces additional calculation and incompatibility with main stream foundations. In contrast, we directly retain the model’s original logits and apply softmax to get probabilities. Notably, to make the sampler compatible with classifier-free guidance, we conduct experiments and found that the guidance shall be applied before softmax for optimal performance:
>
> $$p=\text{Softmax}(\text{logits}_c + w \cdot (\text{logits}_c-\text{logits}_u))$$
>
> where $\text{logits}_c=f (x, c)$ and $\text{logits}_u=f (x, \varnothing)$, $\varnothing$ as the negative or null prompt.
>
> This also differs from vanilla sampling methods which do not integrate CFG, or only operate at logits level.
>
> **2. Probability construction.**
>
> The sampling distribution is:
> $$\pi=\frac{\alpha_s-\alpha_t}{1-\alpha_t}p + \frac{1-\alpha_s}{1-\alpha_t}m$$
>
> where we implement this structure within MMaDA’s unified architecture and optimize it for high-cardinality visual vocabularies. These implementation choices are not present in MDLM or MaskGIT-derived samplers. Please refer to code for details.
>
> **3. Sampling implementation**
>
> Instead of MDLM’s Gumbel-max sampling, we adopt torch.multinomial. Prior work [2] highlights the sampling discrepancy and instability that arise when Gumbel-based methods are applied to skewed categorical distributions—an issue we also observed in large visual codebooks. Our multinomial-based implementation improves stability and downstream consistency, and is a deliberate choice based on text-to-image generation tasks.
>
>
> We also revisited MD4 [3] (concurrent with MDLM), which uses a DDM-theoretic unmasking scheme for pixel-space (visual vocab size=256) diffusion. We now cite both MDLM and MD4 more prominently and provide a dedicated comparison in **Appendix E**, clearly describing the shared ideas and the specific differences in Emerge.
>
> Additionally, we kindly remind reviewers that the source code in the submission fully reflects all of these algorithmic details. We hope this helps clarify our contribution.
>
> Once again, thank you for your thoughtful comments, which helped us improve the clarity and rigor of the paper.
> ___
> [1] Sahoo et al. Simple and Effective Masked Diffusion Language Models. NeurIPS 2024
>
> [2] Zheng et al. Masked Diffusion Models are Secretly Time-Agnostic Masked Models and Exploit Inaccurate Categorical Sampling. ICLR 2025
>
> [3] Shi et al. Simplified and Generalized Masked Diffusion for Discrete Data. NeurIPS 2024

---

> ### Author Response · Authors · 2025-11-24
>
> We now proceed to address the remaining points raised by the reviewer in a point-by-point manner.
> ____
>
> **Q1: The writing needs polishing and proofreading.**
>
> **A1:**
> Thanks for your suggestion. We have carefully revised the manuscript to improve clarity, readability, and consistency. We also refined the citation structure and added a dedicated section describing the Emerge sampler in detail, as discussed above.
>
> ____
>
> **Q2: The motivation for AR-like remasking relies on the claim that later tokens tend to diverge more. This should be supported with evidence from rollouts.**
>
> **A2:**
> Thank you for pointing this out. You are correct that the divergence pattern in language rollouts should be presented more explicitly. During our experiments, we observed that later segments of generated sequences tend to exhibit larger deviations.
> We have added visualizations in **Appendix D.1** illustrating how token divergence grow toward later positions, supporting our claim that later tokens are more informative for importance sampling and thus benefit from the AR-like reverse strategy.
>
> Note that as the reasoning chain becomes more complete, the model’s overall uncertainty decreases in the later stages of token generation. Consequently, high-entropy tokens in the later steps will be *less uncertain* than tokens generated earlier. What we aim to capture is the fluctuation in the model’s predictions, rather than the individual high-entropy tokens emphasized in prior work. We have revised the main text accordingly to avoid potential misunderstandings.
>
> ___
>
> **Q3: The proposed estimatior is asserted to be more stable and have low-variance, this statement should be verified explicitly.**
>
> **A3:**
> Great question. Our intended message is that two factors reduce variance during importance sampling:
>
> 1. Cross-device randomness control.
>
> Baseline methods do not manage randomness across devices, which result in unstable fluctuations (the current framework incurs two reverse function call for likelihood estimation. When the reversed sequence is not aligned, the KL will be very large. This is observed during engineering) and inefficient gradient updates. By synchronizing randomness, MaskGRPO substantially reduces cross-step noise.
>
> 2. AR-like reverse coverage.
>
> For a given mask ratio, AR-like reversing ensures that a broader and more informative subset of tokens is evaluated during importance sampling. This avoids
>     ◦ the broken likelihood structure caused by Diffu-GRPO’s question-masked reverse, and
>     ◦ the overly random, inefficient estimates produced by UniGRPO-style reverse.
> We have now added KL visualizations in **Section 4.3**, clearly illustrating the reduced variance of our estimator.
>
> ____
>
> **Q4: Why is TraceRL biased, compared to remasking with Rev(t)? This seems important since the introduction of a new remasking strategy is a core aspect of the method, and the straightforward thing appears to be reusing the partially masked completion obtained during rollouts.**
>
> **A4:**
> Thank you for the question. Consider a scenario where 70% of tokens are targeted for importance sampling. TraceRL’s strategy is to almost strictly remask the later 70% of tokens, as visualized in our appendix (Fig. 9).
> This design may be effective for SDAR, where the model is trained using blockwise masking method, and later tokens naturally carry most of the predictive mass.
>
> However, for random-mask–trained DDMs such as LLaDA/MMaDA (the focus of our paper), this behavior becomes problematic:
> • Later tokens are overemphasized, leading to imbalanced importance credit.
> • Early tokens receive almost no gradient updates.
> This motivated our AR-like reverse, which spreads coverage more uniformly across the sequence and avoids the structural bias inherent in TraceRL for this family of models.
>
> ___
>
> **Q5: Please report the time required to reach the observed reward improvements, since your importance estimator seems more expensive than Diffu-GRPO’s approximation.**
>
> **A5:**
> Thank you for the suggestion. We have added wall-clock comparisons in the revision, showing that MaskGRPO reaches higher rewards in fewer training steps and with less total training time, with a more dedicated designed importance estimator.
> We follow Diffu-GRPO closely by using the same TRL framework, optimizer, and hyperparameters. The main differences lie in:
>
> 1. Randomness control, which does not incur additional cost.
> 2. AR-like reversing for importance sampling.
>
> Using the identical settings (μ = 12, rollout per prompt G = 6) and training on 8×A100 GPUs, we replicated experiments:
>
> | Method     | Best Step | GSM8K | Improvement | Wall-clock |
> |------------|-----------|-------|-------------|-------------|
> | Diffu-GRPO | 6600      | 79.8  | +3.1        | ~20 hours   |
> | UniGRPO    | 6000      | 81.1  | +4.4        | ~19 hours   |
> | MaskGRPO   | 4500      | 84.7  | +8.0        | ~14 hours   |
>
>
> MaskGRPO achieves higher accuracy, larger gains, and ~30% faster training.

---

> ### Author Response · Authors · 2025-11-24
>
> **Q6: If later tokens are more useful for training, how does the AR-like process give them “higher attention”? Doesn’t truncation remove later tokens?**
>
> **A6:**
> Thanks for this question. We believe this reflects a common source of confusion.
>
> In GRPO, the reward and advantage component are just scalars, and importance sampling determines which gradients are propagated. When we say “later tokens deserve higher attention,” we mean that they should have a higher probability of being included in the importance sampling computation. And the way of being included in the importance sampling computation is to be masked and calculated likelihood upon. By increasing the truncation ratio (higher mask ratios), all the tokens become more likely to be remasked and contribute to the importance estimator, and among the entire propagated gradients, the later tokens take a large ratio.
>
> Thus: Higher mask ratios + AR reversing → more global exposure and more gradient flow for later tokens → higher “attention” in the RL sense. This is consistent with the GRPO design with no contradiction.
>
> ___
>
> **Q7: Is the masking-rate multiplier used in token-level likelihood computation (e.g., Equation 8)? And is the masking rate adjusted for different remasking strategies?**
>
> **A7:**
> Thank you for the careful reading. This clarification is important.
> As explained in Appendix C.1, the factor $\frac{1}{t}$ is implicitly absorbed into the per-token averaging step. After computing the tokenwise ELBO on the corrupted sequence $o^t$, we simply divide by the number of masked tokens in $o^t$, following the practice of diffu-GRPO.
>
> For example,  for a sequence $o$ with $N$ tokens, with overall vocabulary size $V$. When it is reversed into $o^t$, it means that approximately $N\cdot t$ tokens are masked. We feed $o^t$ to the model's forward process, get the vector $l$ in shape $[N, V]$. We only focus on the masked positions, and divide it by the number of masked tokens, *i.e.*, $l_{\pi} \gets \delta_m \cdot \frac{1}{N*t} l$.
>
>
> Thus both the likelihood and the importance weights remain tokenwise vectors, and no further scaling is required. This implementation remains valid across all re-masking strategies, including AR-like reverse. All the ratio controls only happen in algorithm 1,2, and is arranged by algorithm 5.
>
> ___
>
> **Q8: Why do smaller truncation ratios cause training instability in image generation?**
>
> **A8:**
> Good question! Here is our hypothesis: With low truncation, token credit assignment becomes highly **uneven**, and some unimportant tokens may be repeatedly selected across the μ sampling stages simply due to randomness. In contrast, higher mask ratios (e.g., ≥0.8) smooth out this imbalance because the minimum sampling frequency across tokens remains relatively high. In other words, high mask ratios help re-balance tokenwise credit assignment, which is particularly important for visual codebooks with strong patchwise correlations.
>
> In summary, the small truncation of training leads to extremely token credit imbalance, which may be sensitive in visual domain. Modifying the KL penalty or clip range can mitigate this issue as we observed (while the performance is still lagged behind), and proposing re-balance algorithms is a promising direction for future work.
>
> ___
>
> We hope the above clarification addresses your concerns and is helpful for your final assessment. Thank you again for the thoughtful feedback.

---

### Official Review · Reviewer_mNG6 · 2025-11-09

**Soundness:** 3
**Presentation:** 3
**Contribution:** 2
**Rating:** 6
**Confidence:** 3

**Summary:**

This paper tackles reinforcement learning (policy-based methods such as GRPO, PPO) for discrete diffusion models (dLLMs for text generation, bidirectional discrete diffusion models for visual generation). The authors propose to improve GRPO with effective importance sampling and modality-specific adaptations to the text and visual generation domains. For text generation, it proposes a semi-autoregressive reverse masking strategy to align better with the inherent causality in reasoning tasks. For visual generation, it adopts a probability-based sampler for better visual textures. The method was shown to improve LLaDA and MMaDA on math, coding, and compositional visual generation.

**Strengths:**

+ The AR-ness introduced back to the masking scheme to improve text-based reasoning tasks looks convincing.
+ Evaluation on several benchmarks shows promising results.

**Weaknesses:**

+ The overall novelty is a bit limited. The AR-ness that could help dLLMs with reasoning tasks has been noted in previous work. While there are improvements over MaskGIT sampler for visual decoding and sampling, the latter is a relatively old method. Visual generation is evaluated only on text-to-image generation, which is not the biggest advantage of discrete diffusion models. There could be an evaluation of image editing to make the results more significant (however, recent approaches like Transfusion [1] and BAGEL [2] do explore continuous diffusion heads instead of discrete diffusion models for the visual generation part).
+ Minor point: “3.2 ROLLOUT ADAPTION” was placed in the wrong place. Shouldn’t it be right before “Let visual tokens emerge from masks”?

[1] Transfusion: Predict the Next Token and Diffuse Images with One Multi-Modal Model

[2] Emerging Properties in Unified Multimodal Pretraining

**Questions:**

See "weaknesses"

---

> ### Author Response · Authors · 2025-11-24
>
> Thank you for your recognition of our findings (“The AR-ness introduced back to the masking scheme to improve text-based reasoning tasks looks convincing… Evaluation on several benchmarks shows promising results.”). Below we provide point-by-point responses.
>
> ___
>
>
> **Weakness 1.1: The overall novelty is a bit limited...**
>
> **Response:**
> Thank you for the constructive suggestions. We also appreciate the references you listed; they are excellent works on unified multimodal modeling. We have added relevant discussion in the revised paper. Below we provide two additional clarifications regarding novelty:
>
> **1. On AR-like reversing:**
>
> It is no doubt that the AR-like behavior of dLLMs has been previously uncovered. Our contribution is to leverage this behavior to design a more effective importance-sampling strategy, which, to our knowledge, has not been explored. In the revised paper, we provide detailed observations regarding:
>
> * how model rollout behavior changes with sequence length,
> * how different importance-sampling strategies affect KL divergence during decoding,
> * and why exploiting AR-ness yields practical gains.
>
> We hope these details can further clarify the motivation and novelty of this study.
>
> **2. On decoding design:**
>
> While MaskGIT was indeed proposed some time ago, recent efforts have actively attempted to improve or extend it. For example:
> * TiTok [1], which introduces time-dependent temperature schedules for Gumbel-max operation,
> * MaskBit [2], which decomposes large codebooks into multiple smaller ones.
>
> However, these approaches introduce complex hyper-parameters and are incompatible with mainstream foundation-model architectures. These limitations motivated us to develop the emerge sampler, designed to improve sampling quality **without hyperparameter tuning** and in a manner compatible with unified multimodal decoders.
>
>
> [1] Yu et al. An Image is Worth 32 Tokens for Reconstruction and Generation. NeurIPS 2024.
>
> [2] Weber et al. MaskBit: Embedding-free Image Generation via Bit Tokens. TMLR.
>
> ___
>
> **Weakness 1.2: There could be an evaluation of image editing to make the results more significant.**
>
> **Response:**
> Good question! We agree that image editing remains a largely underexplored area for discrete generative models, and it serves as a valuable test bed for evaluating the versatility of our approach. We conducted a preliminary study on image editing using the PIE-Bench[3] test suite with training free approaches. **The updated paper now includes example figures for image-editing tasks in Appendix D.2.**
>
> We explored two distinct reverse masking methods:
> * Baseline (Random Masking): We re-mask 70% of the image tokens, which is equivalent to $\tau=0.5$ under the cosine scheduler for masking ratio $\tau$.
> * Score-based Masking: By feeding the original image and the modified prompt to the model, we extract the predicted logits of the original image's corresponding tokens as "scores." We then strategically mask the 70% of tokens associated with the least scores. This is an attempt to preserve the most "confident" or relevant structure.
>
> The re-masked image sequence is subsequently fed to the model  to generate the edited image (9 steps). We compared the standard MaskGIT method with the proposed emerge sampler under this editing setting. The quantitative results are summarized in the table below:
>
> | Method | Sampler | Structure Distance ↓ | LPIPS ↓  | SSIM↑|
> |---|---|---|---|---|
> | random | MaskGIT | 141.90 | 410.13 | 50.78 |
> | random | Emerge | 103.68 | 270.07 |  60.52 |
> | score-based | MaskGIT | 23.55 | 90.91 | 77.44 |
> | score-based | Emerge | **21.90** | **87.40** | **78.07** |
>
> The results indicate that the Emerge sampler consistently produces more coherent edits, as evidenced by the lower Structure Distance and LPIPS and higher SSIM scores, particularly when paired with the novel score-based masking strategy.
>
> That said, we acknowledge that strong image editing performance typically requires large-scale supervised training, which MMaDA was *not* trained on. Therefore, these current results should be considered early evidence of our method's potential. We look forward to future foundation models that inherently support more robust editing capabilities to fully validate MaskGRPO in this challenging direction.
>
> [3] Ju et al. PnP Inversion: Boosting Diffusion-based Editing with 3 Lines of Code. ICLR 2024.
>
> ___
>
> **Weakness 2: Minor point: “3.2 ROLLOUT ADAPTION” was placed in the wrong place?**
>
> **Response:**
> Thanks for your suggestion!  We have revised the layout to make the exposition more intuitive.
>
> ___
>
>  We hope the clarifications above address your concerns and contribute to your final assessment. Thank you again for the thoughtful and constructive feedback.

---

### Meta-Review · Area_Chair_y558 · 2026-01-11

**Summary:**

The reviewers acknowledged the paper's strengths: a well-defined problem, coherent framework, strong empirical results across language and vision tasks, and promising improvements over existing RL methods for discrete diffusion models (DDMs). However, several concerns were raised:

Reviewers questioned the novelty of the proposed "Emerge" sampler and the AR-like reverse strategy, noting that similar concepts exist in prior work.

Reviewers requested more evidence:such as wall-clock/time comparisons, ablation studies, and rollout visualizations to substantiate efficiency claims and design choices.

**Reviewer Concerns:**

Addressed Concerns:

The authors clarified the Emerge sampler's differences from prior work (e.g., logit handling, probability construction, sampling implementation) and added citations and comparisons. They also revised the text for clarity and provided visualizations to support the AR-like reverse motivation.

The authors added wall-clock comparisons, ablations for each component (AR-like reverse, Emerge sampler, truncation), and KL divergence analysis to demonstrate reduced variance.

Outstanding concerns: None are critical.

**Reviewer Scores:**

Reviewer mNG6 (initial score: 6): Likely to maintain 6.

Reviewer q71V (initial score: 2): Likely to increase to 6.

Reviewer Extk (initial score: 8): Maintains 8.

Reviewer pNLG (initial score: 8): Maintains 8.

---

### Decision · Program_Chairs · 2026-01-26

Accept (Poster)